# Towards Fine-grained Audio Captioning with Multimodal Contextual Fusion

## Abstract

High-quality, large-scale audio captioning is crucial for advancing audio understanding, yet current automated methods often generate captions that lack fine-grained detail and contextual accuracy, primarily due to their reliance on limited unimodal or superficial multimodal information. Drawing inspiration from human auditory perception, which adeptly integrates cross-modal cues and performs sophisticated auditory scene analysis, we introduce a novel two-stage automated pipeline. This pipeline first employs specialized pretrained models to extract diverse contextual cues (e.g., speech, music, general sounds, and visual information from associated video). A large language model (LLM) then synthesizes these rich, multimodal inputs to generate detailed and context-aware audio captions. Key contributions of this work include: (1) the proposed scalable method for fine-grained audio caption generation; (2) FusionAudio, a new large-scale dataset comprising 1.2 million such detailed captions, combined with 6 million QA pairs; and (3) enhanced audio models developed using FusionAudio, specifically a CLAP-based audio encoder with superior audio-text alignment and instruction following. This paper paves the way for more nuanced and accurate automated understanding of complex audio environments.

## 1 Introduction

The advancement of models like CLAP Wu* et al. (2023) for audio retrieval, and GAMA Ghosh et al. (2024) or Qwen2-Audio Chu et al. (2024) for broader audio understanding, heavily relies on large-scale, high-quality audio captioning datasets. Audio captioning has primarily followed two trajectories. *Manual annotation* Drossos et al. (2019); Kim et al. (2019) offers high quality but lacks scalability due to high labor costs. In contrast, *automated methods* LAION-AI (2023); Mei et al. (2024) often use sparse metadata like text labels or tags to assist annotation, while others Bai et al. (2024); Sun et al. (2024); Yuan et al. (2025) leverage basic multimodal cues. These automated approaches, however, typically rely on limited textual or superficial information, failing to capture rich details (e.g., multimodal contextual details). This results in captions that lack fine-grained details and are prone to hallucinations Yang et al. (2024), hindering nuanced audio interpretation.

Addressing this gap necessitates a paradigm shift. We turn to human auditory perception for inspiration (Figure 1). Human auditory understanding leverages sophisticated strategies at two complementary levels. **Firstly**, humans adeptly integrate cross-modal cues—visual information, for instance, aids speech intelligibility Sumby & Pollack (1954) and sound identification Kayser et al. (2010); Ernst & Bülthoff (2004). **Secondly**, auditory scene analysis (ASA) Bregman (1990) allows the auditory system to parse complex soundscapes into distinct streams like speech, music, and ambient sounds based on temporal-spectral regularities Shamma et al. (2011). These sophisticated biological mechanisms offer a compelling blueprint for enhancing automated audio captioning.

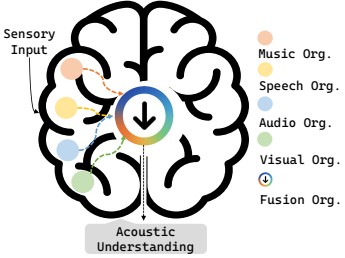

Figure 1: Human auditory perception integrates multisensory cues.

The impact of this multimodal integration is demonstrated in Table 1. Current systems, often processing audio in isolation, can misinterpret sounds (e.g., a stationary motorcycle as a moving scooter) or hallucinate details. In contrast, FusionAudio-1.2M leverages comprehensive audiovisual cues to produce more accurate and contextually rich descriptions.

Table 1: Comparison of generated captions for a sample audio clip with associated visual context. Hallucinations in prior work are highlighted in red. Improvements from our multimodal approach, FusionAudio, are highlighted in green, demonstrating enhanced accuracy and detail by leveraging visual and comprehensive auditory cues.

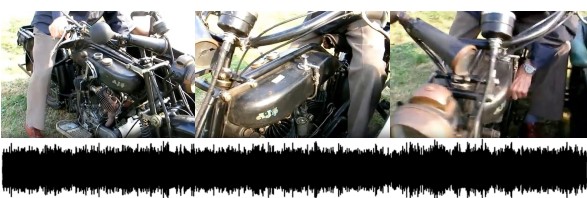

| Method | Audio Caption |
|---|---|
| **GAMA (Baseline)** | The audio is dominated by the sound of a motor vehicle engine and intermittent male speech, with wind noise. |
| **AudioSetCaps** | A male and female engage in conversation, their voices audible against a backdrop of ambient noise. The discussion is neutral in tone and does not involve any identifiable objects or language. |
| **Auto-ACD** | A man speaks while a vehicle moves in the distance, possibly on a motor scooter, in an engine room. |
| **Sound-VECaps** | A man is speaking and a vintage motorcycle with a large headlamp, round fuel tank, and sidecar is parked on grass, with the sound of the engine and the man's voice filling the air, while a vehicle passes by in the background. |
| **FusionAudio-1.2M (Ours)** | Continuous motor vehicle engine noise is prominently featured, accompanied by intermittent male speech with a positive or confirming tone. Wind sounds suggest an outdoor environment, with the engine's sustained roar maintaining a steady volume throughout the recording. |

Inspired by these principles, we introduce a two-stage pipeline for enhanced automated audio captioning. First, specialized pretrained models extract diverse contextual cues: an Automatic Speech Recognition (ASR) model Radford et al. (2022) for speech, a music understanding model Zhao et al. (2024) for musical attributes, an audio understanding model Ghosh et al. (2024) for general sounds, and a visual model Bai et al. (2025) for video information. Second, a large language model (LLM) Team (2025) acts as an integration engine, synthesizing these multimodal cues to generate fine-grained audio captions. This synthesis of rich, cross-modal context by an LLM aims to improve detail and accuracy, addressing prior limitations.

Our contributions are:

- Automated fine-grained audio captioning: A pipeline using specialized unimodal models to extract diverse contextual cues, synthesized by an LLM to generate detailed, scalable captions.

- FusionAudio-1.2M dataset: A large-scale dataset of 1.2M fine-grained audio captions to advance audio research.

- Multimodal cue-enhanced audio models: A CLAP-based audio encoder with improved audio-text alignment, and an instruction-tuned MLLM with stronger audio comprehension and instruction-following.

## 2 MOTIVATION:HOW HUMANS PERCEIVE AND COMPREHEND SOUND

### 2.1 MECHANISM OF MULTIMODAL-ASSISTED AUDITORY COMPREHENSION

A large body of neuroscientific and biological research has explored or demonstrated the role of cross-modal synergy in auditory comprehension. In addition to the aforementioned studies (e.g., Sumby & Pollack (1954)),McGurk & MacDonald (1976) identifies the McGurk Effect—key evidence that the brain efficiently integrates multimodal information. This phenomenon shows human auditory perception is not isolated; instead, it interacts deeply with senses like vision to construct coherent

Table 2: Comparison of open-source audio caption datasets.

| Name | Year | # of Audio/QA | Avg. Dur (s) | Avg. Text Len | Visual | Music | Speech | Integration |
|------|------|---------------|--------------|---------------|--------|-------|--------|-------------|
| AudioCaps Kim et al. (2019) | 2019 | 46k/46k | 10.00 | 9.03 | ✗ | ✗ | ✗ | ✗ |
| Clotho Drossos et al. (2019) | 2019 | 5k/5k | 22.50 | 11.00 | ✗ | ✗ | ✗ | ✗ |
| LAION-Audio-630K LAION-AI (2023) | 2022 | 630k/630k | 24.58 | 7.30 | ✗ | ✗ | ✗ | ✗ |
| WavCaps Mei et al. (2024) | 2024 | 403k/403k | 67.59 | 7.80 | ✗ | ✗ | ✗ | ✗ |
| AudioSetCaps Bai et al. (2024) | 2024 | 1.9M/1.9M | N/A | 28.00 | ✗ | ✗ | ✗ | ✗ |
| Auto-ACD Sun et al. (2024) | 2024 | 1.5M/1.5M | 10.00 | 18.10 | ✓ | ✗ | ✗ | ✓ |
| CompA-R Ghosh et al. (2024) | 2024 | 62k/200k | 9.93 | 18.00 | ✓ | ✗ | ✗ | ✓ |
| **FusionAudio-1.2M (Ours)** | **2025** | **1.2M/6M** | **10.00** | **47.18** | ✓ | ✓ | ✓ | ✓ |

external cognition.Wei et al. (2022) further elaborates on how multimodal information aids auditory comprehension: light is received by retinal photoreceptors and sound converted to neuronal signals at the eardrum. After separate processing of audio and visual information, advanced nervous systems (e.g., superior colliculus, superior temporal sulcus) handle cross-modal input, while many neurons process fused multisensory signals—yielding more reliable responses than unimodal ones. The cerebral cortex also forms a module for coordinated multisensory integration to build consciousness and cognition.These findings offer valuable insights for the design of our pipeline.

## 2.2 PIPELINE FOR BIONIC AUDITORY COMPREHENSION MECHANISM

Based on existing research, in-depth audio comprehension involves not only the auditory and higher neural centers but also intricate interactions with other nervous systems (e.g., visual center, language-comprehension brain regions). To mimic this biological mechanism and improve multimodal models' audio comprehension, we design a two-stage pipeline:The first stage uses different multimodal models to generate respective understandings of audio or its corresponding video—mirroring how organisms collect information from multiple senses (e.g., vision, hearing);the second stage employs an overall model to organize, summarize cross-modal information, and produce the final auditory comprehension—aligning with the multimodal information synergy mechanism in organisms' advanced neural centers.This pipeline is expected to help large models better match biological auditory comprehension mechanisms, thereby enhancing audio comprehension effectiveness.The results of comparing the datasets generated by our pipeline with others are presented in Table 2.

## 3 METHOD: FINE-GRAINED AUDIO CAPTION WITH MULTIMODAL CONTEXTUAL FUSION

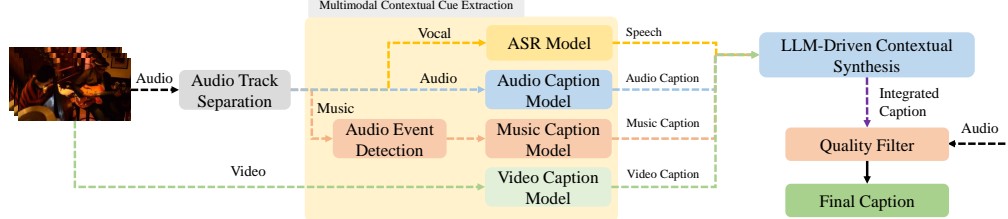

Figure 2: Overview of our proposed multimodal audio captioning pipeline. The process involves initial vocal separation, followed by a two-stage approach: multimodal contextual cue extraction and LLM-driven contextual synthesis.

### 3.1 AUTOMATED CAPTIONING PIPELINE

We introduce a two-stage pipeline, illustrated in Figure 2, designed to generate fine-grained audio captions: (1) Multimodal Contextual Cue Extraction using specialized expert models, and (2) LLM-Driven Contextual Synthesis to integrate these diverse cues into a coherent caption.An initial pre-processing step is performed to enhance audio quality.

**Pre-processing: Audio Track Separation.**  To improve the quality of the speech transcription, we first apply Demucs model Rouard et al. (2023) to isolate the vocal track from the audio stream.

**Stage 1: Multimodal Contextual Cue Extraction.** This stage leverages a suite of specialized models to extract diverse, complementary information streams relevant to the auditory scene. The prompts used for these models can be found in Appendix C.

- **General Audio Events:** We utilize GAMA Ghosh et al. (2024) to generate captions to capture overall acoustic scene characteristics(sound events and environments).
- **Speech Content:** Separated vocal is transcribed by Whisper model Radford et al. (2022).
- **Music Characteristics:** We first use YamNet TensorFlow (n.d.) to confirm music presence, mitigating hallucination risk on non-musical segments. If music is detected, OpenMu Zhao et al. (2024) is used to extract details regarding genre, instrumentation, tempo, and mood.
- **Visually-Grounded Context:** We utilize Qwen2.5-VL-72B Bai et al. (2025) to extract visual information from the video stream. This approach yields a detailed, timestamped visual record, providing visual context that aids in grounding physical events.

**Stage 2: LLM-Driven Contextual Synthesis.** Extracted info streams feed into the synthesis model QwQ-32B Team (2025). As an integration engine, the LLM is prompted to: (a) coherently synthesize multimodal inputs, (b) resolve redundancies or minor inconsistencies across expert outputs, (c) infer relationships and context implied by combined information, (d) generate a final fine-grained audio caption reflecting understanding of the auditory scene enriched by multimodal context.

## 3.2 Data Source

To demonstrate the effectiveness of the pipeline,we use it to generate captions for AudioSet Gemmeke et al. (2017),which provides over 2 million 10-second YouTube corresponding audio and video clips.

## 3.3 Data Quality Assurance

To ensure the quality and reliability of generated captions, we adopt a multi-faceted quality assurance protocol, encompassing manual verification on sample data (more details are in Appendix B) and scalable automated filtering for curating the final FusionAudio-1.2M dataset.

**Manual Verification.** To establish a benchmark for caption quality, we randomly sampled 1300 generated captions for human evaluation. Trained annotators assessed each caption based on two criteria: **(1) Detailness:** Rated on a 3-point scale, a higher score means more details, evaluating the richness and specificity of the information conveyed. **(2) Hallucination:** Rated on a 5-point scale, a higher score means less hallucination, assessing the factual accuracy of the caption against the audio-visual content. A score of $\leq 2$ is considered indicative of notable hallucination.

As shown in Table 3, the manually evaluated sample achieved a mean detailness score of 2.48/3. For hallucination, the average score is 3.98, with 6.2% of captions scoring of 2 or lower (indicating low significant inaccuracies). Inter-annotator agreement (calculated via exact match rate) is 0.59 for detailness and 0.91 for hallucination—these values indicate moderate agreement, which is reasonable given the subjectivity of fine-grained caption quality assessment.

**Automatic Filtering** For quality assessment, we filter data with obvious audio-text mismatch by using cosine similarity between audio and caption embeddings from CLAP as quality indicator. We label hallucination scores $\leq 2$ as positive class (captions to discard) and scores $> 2$ as negative class (captions to retain) by annotators as groundtruth, then evaluate various cosine similarity thresholds using $F_{1.05}$ score, which slightly emphasizes recall to prioritize removing hallucinated content. Threshold 0.08, which is used for filtering aligns optimally with human(exact match rate: 88.3%), yielding a 7.3% filter rate (More details in Appendix B.3).

Table 3: Manual Verification Results. Detailness is rated 1-3 (higher is better). Hallucination is rated 1-5 (higher is better; $\leq 2$ indicates notable hallucination). IAA is measured using the exact matching, before which hallucination score has been converted to 1 (score $\leq 2$) or 0 (score $> 2$).

| Caption Content Quality | | Inter-Annotator Agreement | |
|---|---|---|---|
| Detailness | Hallucination | Detailness | Hallucination |
| 2.48 | 3.98 | 0.59 | 0.91 |

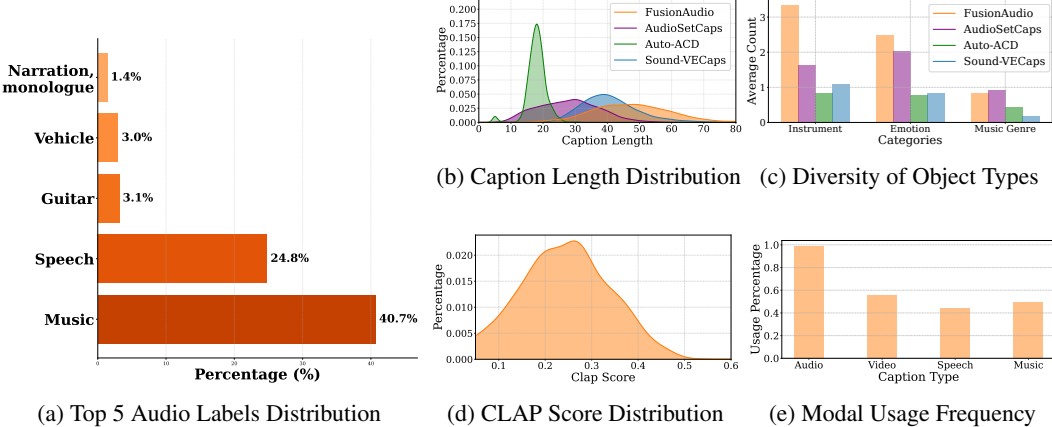

(b) Caption Length Distribution  (c) Diversity of Object Types

(a) Top 5 Audio Labels Distribution  (d) CLAP Score Distribution  (e) Modal Usage Frequency

Figure 3: Key statistics of FusionAudio-1.2M: (a) Proportion of top 5 audio labels from AudioSet; (b) Caption length comparison with existing datasets; (c) Diversity of semantic content types; (d) Distribution of audio-text similarity from CLAP; (e) Proportion of modalities in captions.

## 4 THE RESULTED DATASET: FUSIONAUDIO-1.2M

### 4.1 QUANTITATIVE ANALYSIS

Table 2 compares our proposed dataset with other publicly available datasets. FusionAudio-1.2M distinguishes itself through its large scale, longer caption length, and integration of multiple modalities.

**Dataset Statistics** We analyze FusionAudio-1.2M across several dimensions:

- **Audio Category Distribution:** Figure 3a shows top 5 highest proportion of the occurrence count of each different audio label in all audio clips relative to the total number of audio clips AudioSet Gemmeke et al. (2017). The sum of all proportions is greater than 1.

- **Caption Length:** Figure 3b compares caption lengths (in words) with AudioCaps Kim et al. (2019), Sound-VECaps Yuan et al. (2025) and Auto-ACD Sun et al. (2024). FusionAudio-1.2M captions are significantly longer, indicating greater descriptive richness.

- **Semantic Diversity:** For comparing semantic richness across datasets,we use GPT-4o-mini (prompts in Appendix C.4) to identify *instruments*, *emotions*, and *music genres* in each caption . Figure 3c shows FusionAudio-1.2M has higher coverage across most categories.

- **Audio-Text Alignment:** Figure 3d shows the distribution of cosine similarity between audio and text embeddings calculated by CLAP Wu* et al. (2023). Samples of different similarity scores can be found in Appendix C.7.

- **Modality Usage:** GPT-4o-mini is used to annotate captions for modalities: *audio events*, *speech*, *music*, *visual context* (see prompt in Appendix C.4) to know how modalities contribute to captions. Figure 3e shows over 50% of samples integrate at least 2 modalitie.

### 4.2 QUALITATIVE ANALYSIS

**Case Study** Table 1 compare caption for the same audio clip across datasets. FusionAudio's caption integrates multi-dimensional cues, and shows detail and reasoning absent from prior datasets.

**Embedding Projection for Visualizing Semantic Granularity** Embedding projection techniques like t-SNE van der Maaten & Hinton (2008) visually reveal a dataset's semantic structure(intraclass compactness, inter-class separability)—key for discriminative task data quality.We apply it to project CLAP sentence embeddings of FusionAudio-1.2M and other datasets' captions. Figure 4 demonstrates that FusionAudio's captions form far more compact same-category clusters and clearer inter-category separation than baselines. This confirms its superior semantic granularity and discriminative power. Quantitative validation of inter- and intra-class distances is in Appendix D.1.

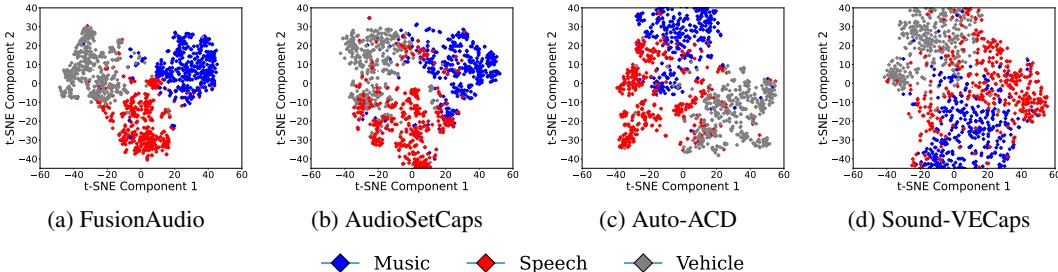

|  |  |  |  |
|---|---|---|---|
| (a) FusionAudio | (b) AudioSetCaps | (c) Auto-ACD | (d) Sound-VECaps |

◆ Music  ◆ Speech  ◆ Vehicle

Figure 4: T-SNE Embedding of popular categories between different datasets

## 5 APPLICATIONS OF FUSIONAUDIO-1.2M

FusionAudio-1.2M is used for two downstream tasks:audio-text retrieval(Sec. 5.1) and audio understanding ( Sec. 5.2).Experiments run on a server with 8 NVIDIA A800 80GB GPUs.

### 5.1 AUDIO-TEXT RETRIEVAL

Table 4: Audio-text retrieval performance (R@k, %) on the AudioCaps test set.

| Dataset | Model | Text - to - Audio | | | Audio - to - Text | | |
|---|---|---|---|---|---|---|---|
| | | R@1 | R@5 | R@10 | R@1 | R@5 | R@10 |
| AC+CL | HTSAT+BERT | 36.1 | 71.8 | 83.9 | 46.8 | 82.9 | 90.7 |
| WavCaps | HTSAT+BERT | 42.2 | 76.5 | 87.1 | 54.6 | 85.2 | 92.4 |
| AudioSetCaps | HTSAT+BERT | 43.4 | 78.4 | 88.2 | 57.3 | 84.2 | 93.2 |
| Auto-ACD | HTSAT+RoBERTa | 42.7 | - | 88.5 | 56.3 | - | 93.9 |
| Sound-VECaps | HTSAT+RoBERTa | 39.2 | 74.1 | 85.0 | 54.0 | 82.5 | 93.2 |
| FA(Ours) | HTSAT + BERT | **44.3** | **79.9** | **90.4** | **57.8** | **86.1** | **94.4** |

#### 5.1.1 EXPERIMENTAL SETUP

**Tasks and Models**    We evaluate FusionAudio-1.2M by using it as a pre-training corpus for cross-modal audio-text retrieval. This task requires retrieving the most relevant audio clip for a given caption (text-to-audio) and identifying the most pertinent text description for a given audio(audio-to-text). We employ the HTSAT Chen et al. (2022)-BERT Devlin et al. (2019) model architecture.

**Two-Stage Training**    Our training methodology for all evaluated datasets, including FusionAudio-1.2M and the baselines, follows a consistent two-stage protocol:

- **Pre-training:** HTSAT-BERT is pre-trained on source datasets (e.g., FusionAudio-1.2M, WavCaps) via contrastive learning, with pre-training parameters: learning rate 5e-5, batch size 196, and 15 epochs.

- **Fine-tuning:** The pre-trained model undergoes full-parameter fine-tuning on the official training split of the AudioCaps (AC) dataset Kim et al. (2019), with a fine-tuning learning rate of 1e-5, batch size of 196, and 20 training epochs.

**Evaluation Setting**    Models after two-stage training protocol is evaluated on the official test set of the AudioCaps dataset Kim et al. (2019) by Recall@k (R@k,k={1, 5, 10}) for text-to-audio and audio-to-text retrieval directions. R@k quantifies the percentage of queries for which the ground-truth item is retrieved within the top-k ranked results.

#### 5.1.2 PERFORMANCE ANALYSIS

As shown in Table 4. Models trained on FusionAudio-1.2M outperform those on baseline datasets in all R@k metrics. It shows that our pipeline accurately capture audio information, enabling the model to distinguish fine-grained details and achieve high-accuracy matching for similar audio.

## 5.2 AUDIO UNDERSTANDING

To validate FusionAudio-1.2M's utility and quality,we evaluate GAMA Ghosh et al. (2024)(fine-tuned on it and other datasets) and general-purpose closed-source models on audio understanding tasks.We further demonstrate FusionAudio's quality by Arena method.(details in Appendix F.1).

Table 5: Performance of GAMA model fine-tuned on FusionAudio against baseline datasets and general-purpose closed-source models across audio understanding evaluation benchmarks.M.J. denotes Model-Judge score (using GPT-4.1-mini).Underlines mark the optimal results of FusionAudio-finetuned GAMA versus closed-source models, while bold text marks those versus open-source models.The three main categories of evaluation tasks align with those in Table 9.

| Dataset | Adverse Acoustic Conditions | | | | | High-Level Semantic Understanding | | | | | | | | Fine-grained Information | | | | |
|---|---|---|---|---|---|---|---|---|---|---|---|---|---|---|---|---|---|---|
| | AS (Acc.) | US$_{8k}$ (mAP) | TAU (mAP) | FSD$_{ns}$ (mAP) | Avg. | Genre (Acc.) | M$_{AQA}$ (Acc.) | Mood (Acc.) | M$_{chat}$ (M.J) | S$_{AQA}$ (Acc.) | S$_{chat}$ (M.J) | AB$_{Sc}$ (M.J) | Avg. | Vocal (Acc.) | Instr (Acc.) | ESC (Acc.) | FSD (mAP) | Avg. |
| *Closed-source Models* | | | | | | | | | | | | | | | | | | |
| gpt-4o-audio | 54.4 | 49.2 | 20.1 | 42.9 | 41.7 | 62.8 | 68.1 | 36.6 | 68.2 | 59.5 | 75.4 | 62.2 | 61.8 | 88.9 | 43.9 | 48.3 | 18.3 | 49.9 |
| gemini-2.5-pro | 64.6 | 56.6 | 20.8 | 45.6 | 46.9 | 73.8 | 66.8 | 46.7 | 69.9 | 65.3 | 74.5 | 70.8 | 66.8 | 92.4 | 62.5 | 48.3 | 19.4 | 55.7 |
| *Open-source Model* | | | | | | | | | | | | | | | | | | |
| GAMA(base) | 48.0 | 56.6 | 23.5 | 81.9 | 52.5 | 42.8 | 44.1 | 28.3 | 45.4 | 50.1 | 58.9 | 56.0 | 46.5 | 63.5 | 68.7 | 68.9 | 45.8 | 61.7 |
| AC+CL | 50.3 | 65.3 | 21.3 | 81.9 | 54.7 | 49.4 | 50.7 | 28.4 | 47.0 | 52.3 | 55.4 | 61.3 | 49.2 | 68.2 | 68.9 | 65.7 | 39.9 | 60.8 |
| WavCaps | 55.4 | 64.5 | 25.0 | 77.6 | 55.6 | 53.4 | 51.6 | 33.2 | 27.7 | 45.1 | 29.7 | 52.7 | 41.9 | 55.4 | 69.7 | 58.8 | 32.4 | 54.1 |
| ASC | 45.4 | 51.3 | 22.3 | 77.8 | 49.2 | 56.0 | 57.6 | 31.6 | 51.3 | 49.9 | 59.5 | 58.5 | 52.1 | 51.8 | 70.5 | 57.7 | 30.5 | 52.6 |
| CompA-R | 56.5 | 63.3 | 22.7 | 83.7 | 56.6 | 60.1 | 54.7 | 33.9 | 47.0 | 56.1 | 58.3 | 60.1 | 52.9 | 63.5 | 68.6 | 62.3 | 38.4 | 58.2 |
| *Our Model* | | | | | | | | | | | | | | | | | | |
| **FA** | 59.0 | 58.8 | 24.4 | 84.6 | 56.7 | 65.1 | 57.6 | 35.7 | 57.1 | 59.1 | 61.5 | 64.5 | 57.4 | 69.0 | 73.6 | 65.5 | 44.5 | 63.0 |
| **FA-high** | 59.7 | 64.0 | 25.1 | 88.2 | 59.3 | 64.2 | 60.0 | 38.3 | 57.9 | 58.4 | 62.3 | 64.0 | 57.9 | 71.0 | 73.9 | 71.3 | 47.4 | 65.9 |

### 5.2.1 EXPERIMENTAL DESIGN

**Tasks and Models**    We focus on general audio understanding beyond speech, employing the GAMA model architecture Ghosh et al. (2024), as our foundation for fine-tuning with a learning rate of 5e-5, a batch size of 128, and 2 training epochs. Evaluation utilizes t=0.1 for inference.

**Training**    GAMA is fine-tuned independently on several datasets(30 minutes per run): FusionAudio-1.2M and its subset FusionAudio-high (top 25k QA pairs selected for quality and diversity), alongside established datasets.We **normalize training data to 25,000 QA pairs** across all datasets. Notably, while baseline datasets usually need 25,000 unique audio clips (one QA pair each) for this volume, FusionAudio-1.2M does it with only 9,000, due to multiple rich QA pairs per clip.

**Evaluation**    Fine-tuned models are evaluated on 15 diverse audio understanding tasks (Table 5) in three scenarios(5 hours per model): (1) robustness to Adverse Acoustic Conditions, (2) proficiency in High-Level Semantic Understanding, (3) acuity in discerning Fine-grained Information.

### 5.2.2 PERFORMANCE ANALYSIS

**Dominant Performance Driven by High-Quality and Efficient Data**    As shown in Table 5,GAMA fine-tuned on FusionAudio (notably FusionAudio-high) outperforms other benchmark-trained models across most 13 tasks (highest average scores overall) and surpasses Gemini-2.5-pro/GPT-4o in Adverse Acoustic Conditions and Fine-grained Information—proving the pipeline's value for nuanced audio understanding. Gemini-2.5-pro, though, leads in high-level semantic understanding (a given for large general-purpose models with rich world knowledge).

## 6 ABLATION STUDY

### 6.1 ON THE EFFECTIVENESS OF MULTIMODAL CUES

To rigorously assess how each component of our method boosts audio information, we do a comprehensive ablation study to: (1) identify the individual importance of auxiliary modalities (Speech, Music, Video) for Sound enhancement; (2) verify the effectiveness of filtering module.All ablation experiments are performed on the same subset from AudioSet with a scale of 25k, using the same

training procedures. FusionAudio-1.2M, which includes all four modalities (Sound, Music, Speech, Video) and the multi-modal fusion quality threshold filtering module, is compared against multiple ablated variants.

**Ablation Results on Fusion**    As shown in Table 6, ablating auxiliary modalities (Music, Video, Speech) generally degraded performance. Removing video captions (w/o Video) causes the most significant decline, underscoring visual context's critical role. Ablating music (w/o Music) and speech (w/o Speech) also reduces performance. An interesting exception is observed for Task 1, where removing speech (w/o Speech) leads to a slight improvement. We attribute this to a combination of potentially poor ASR transcription quality in adverse acoustic conditions, which could introduce detrimental noise, and a possible task focus shift where non-speech acoustic analysis is prioritized, making speech content less critical and potentially diverting optimization from core modalities. Notably, the magnitude of these performance drops (-0.76 for Music, -1.18 for Video, and -0.93 for Speech on average) generally corresponds with the usage of these modalities in our dataset, as illustrated in Figure 3e. This suggests that modalities more frequently leveraged for information contribute more significantly to the overall performance.

**Ablation Results on Filtering**    Removing quality filtering module leads to a significant performance drop across all tasks, highlighting its effectiveness in mitigating issues of obvious audio-text mismatch introduced during the pipeline.See the abaltion result of audio understanding in Appendix E.1.

## 6.2    On the Effectiveness of Data Scaling

To assess the impact of data volume, we conduct scaling experiments for the downstream tasks. This study evaluates performance gains as data size increases, providing insights into model scalability.

**Experiment Setup**    We use nested subsets, starting from 1.25K audio clips. The Audio Understanding task scales to 80k clips (355k QA pairs), while Retrieval utilize up to the full 1.2M clips. Model architectures and training hyperparameters remain consistent with previous experiments.

Table 6: Ablation Study on FusionAudio-25K.Retrieval tasks: Text-to-Audio and Audio-to-Text and Understanding tasks (*Task I*: Adverse Acoustic Conditions; *Task II*: High-level Semantic Understanding; *Task III*: Fine-grained Information).

| Settings | Retrieval Task | | Understanding Task | | | Avg. |
|---|---|---|---|---|---|---|
| | T-A | A-T | *Task I*: AAC | *Task II*: HSU | *Task III*: FI | |
| FusionAudio-1.2M | **39.70** | **49.71** | 56.73 | **57.16** | **63.02** | **53.26** |
| w/o Music | 39.03 | 47.53 | 56.72 | 56.34 | 62.87 | 52.50(-0.76) |
| w/o Video | 38.53 | 48.79 | 55.90 | 56.12 | 61.08 | 52.08(-1.18) |
| w/o Speech | 38.09 | 47.87 | **57.38** | 56.06 | 62.27 | 52.33(-0.93) |
| w/o Filter | 39.45 | 49.14 | 55.30 | 55.25 | 61.35 | 52.10(-1.16) |

**Results**    As shown in Figure 5, for Audio Understanding, scaling from 1.25K to 80K clips improved average performance;Retrieval task saw a consistent Recall@1 increase with more data. These results confirm more data boosts model capabilities and our method's scale and richness.

## 7    Related Works

### 7.1    Audio Language Learning

Audio-language models have advanced greatly in recent years, with research focusing on models that process and understand sounds using natural language as supervision.Early works like CLAP Elizalde et al. (2023) laid the foundation for contrastive learning approaches in audio-language pre-training. subsequent studies explored generative/discriminative objectives(e.g., CTAL Li et al. (2021),FLAP Yeh et al. (2023) with masked modeling) and and multi-task learning(e.g., UniAudio Tian et al. (2023),SpeechX Wang et al. (2024)) to enhance representations and cross-modal alignment.Additionally, integrating large language models (LLMs) with audio processing(e.g., Pengi Deshmukh et al. (2023),Qwen-audio  Chu et al. (2023),Audio Flamingo Kong et al. (2024)) opened new

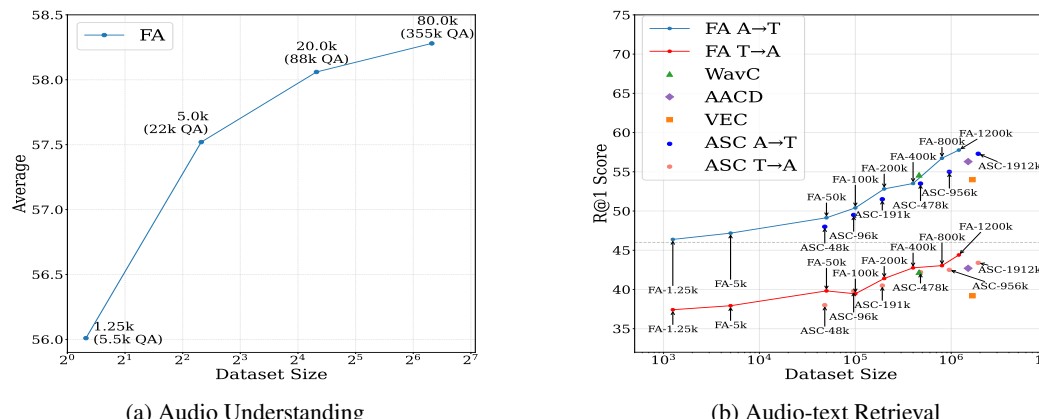

(a) Audio Understanding          (b) Audio-text Retrieval

Figure 5: Scaling result of understanding and retrieval tasks. Details of the legend in (b): A: Audio; T: Text; FA: FusionAudio-1.2M; WavC: WavCaps; AACD: Auto-ACD; VEC: Sound-VECaps; ASC: AudioSetCaps.

avenues for creating more powerful and human-like audio understanding systems.These advancements underscore the models' growing role in linking auditory and language understanding, and potential for real-world applications.

## 7.2 AUDIO CAPTIONING

Early audio captioning research relied on *manually annotated* datasets like AudioCaps Kim et al. (2019) and Clotho Drossos et al. (2019),offering high-quality descriptions but inherently small-scale. To address this, the field turned to **automated and weakly-supervised methods**. These leverage large-scale web-sourced audio with associated sparse metadata (e.g., WavCaps Mei et al. (2024), LAION-Audio-630K LAION-AI (2023)), employ existing textual tags to guide generation, or incorporate basic multimodal cues from loosely associated content Bai et al. (2024); Sun et al. (2024); Yuan et al. (2025).Though boosting scalability, these automated techniques typically yield captions lacking the fine-grained detail and rich contextual understanding characteristic of human annotations or, as we posits, achievable through more sophisticated, deeply integrated multimodal information processing.

## 8 CONCLUSION

This paper presents a novel multimodal contextual fusion pipeline and FusionAudio-1.2M, a large dataset for fine-grained audio captioning. Inspired by human auditory perception, the approach combines expert models (speech, music, sound events, visual context) with LLM-based synthesis. Experiments show that models trained on FusionAudio-1.2M achieve strong performance using fewer unique audio samples due to richer per-clip annotations,which demonstrate the effectiveness of our pipeline.Ablation studies confirm the significance of each modality, particularly visual context. Future improvements to this work include polishing caption generation, try our pipeline on longer audio clips, exploring more advanced multimodal fusion, and deeper societal impact analysis.

## LIMITATION

The study notes several limitations. First, the automated generation of audio captions may introduce hallucinations or errors, even with quality checks (human evaluation, automatic filtering). Second, the dataset used by our pipeline focus on short clips (10s) limits use for longer/more complex audio.Third, multimodal fusion integrates speech, music, visual, and general audio, but modality interplay and weighting are under-explored. Lastly, due to computational resource constraints, we are unable to conduct multiple experimental runs to establish robust error bars for all reported metrics, which could provide further statistical confidence.Future work could address these limitations by further refining the caption generation process, try our pipeline on longer audio clips, exploring more nuanced multimodal fusion strategies, and conducting a more comprehensive analysis of societal impacts.

## ETHICS STATEMENT

All authors of this paper have fully read and strictly adhered to the ICLR Code of Ethics (https://iclr.cc/public/CodeOfEthics), and confirm that the entire research process complies with all requirements of the Code regarding conference participation, paper submission, and academic integrity.

## REPRODICIBILITY STATEMENT

We are committed to ensuring the reproducibility of the research findings reported in this paper, and have structured key supporting information to enable transparent validation. See Chapter 3 for details of our pipeline and Appendix C for prompts the models used.See Chapter 5 for model training details such as hyperparameters and evaluation methods. Code and data samples are provided in supplementary materials.

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

## A  THE USE OF LARGE LANGUAGE MODELS

In this work, we only used Large Language Models (LLMs) for checking grammatical errors, polishing the writing, with no involvement in research ideation or manuscript writing.

## B  HUMAN EVALUATION

### B.1  EVALUATION SETUP

We recruit five evaluators to assess the data. All evaluators are students with a bachelor's degree or higher and have studied in an English-only teaching environment. The five evaluators are tasked with evaluating a total of 1300 samples. Each evaluator is assigned 520 samples, ensuring that each sample is evaluated twice by different evaluators.

Evaluators are required to score the captions based on two dimensions: the level of detailness and the degree of hallucination.

- **Detailness:** Evaluating the level of detail, specificity, and contextual information provided in the caption regarding the audio events and scene. Captions describing multiple relevant aspects accurately scored higher. Detailness is scored through 1-3.
- **Hallucination:** Assessing the accuracy of the description against the source audio-visual content. This specifically penalizes hallucinated objects, events, or attributes not perceivable in the clip. Hallucination is scored through 1-5.

Specific scoring guidelines can be found in the Appendix B.2.

### B.2  INSTRUCTION FOR HUMAN EVALUATION

The instruction used for human evaluation is shown in Figure 6.

### B.3  F-SCORE COMPUTATION

To balance precision and recall in our automatic filtering process, we used the $F_{1.05}$ score, which slightly emphasizes recall over precision. This emphasis ensures that captions with high hallucination rates are effectively discarded, even at the cost of filtering out some acceptable ones. The $F_{1.05}$ score is calculated using the formula:

$$F_{1.05} = \frac{(1 + 1.05^2) \cdot \text{Precision} \cdot \text{Recall}}{(1.05^2 \cdot \text{Precision}) + \text{Recall}}$$

Where precision and recall are computed from the confusion matrix as:

$$\text{Precision} = \frac{\text{TP}}{\text{TP} + \text{FP}} \quad \text{and} \quad \text{Recall} = \frac{\text{TP}}{\text{TP} + \text{FN}}$$

### B.4  HUMAN RATING DISTRIBUTION

We statistically analyze the distribution of human ratings for detailness and hallucination,which are shown as Figure 7.

**Instruction for Human Evaluation**

## INTRODUCTION

You are tasked with evaluating captions generated for audio clips. Please use the following guidelines to assess each caption based on two indicators: **Detailing** and **Hallucinations**

## 1. DETAILING

KEY THINGS TO LOOK FOR:

- Whether the caption captures all major sounds and events in the audio (e.g., dog barking, doorbell ringing, etc.).
- If the intensity or emotional context of the sound is conveyed (e.g., the dog barking intensely or the doorbell ringing in a rapid succession).
- Whether the caption includes additional information when relevant (e.g., a dog barking *repeatedly* or *distressed*).

SCORING GUIDELINES:

Categorize captions into three detail levels (high, medium, low)based on their coverage of audio elements.

- **Low:** Only generic descriptions without specific elements
- **Medium:** Identifies main elements but lacks contextual details
- **High:** Specifies sound sources, qualities, and relationships

## 2. HALLUCINATIONS

You will be given the highlighted words or phrases marked by DeepSeek-V3 that need to be verified in the original caption:

A [**male voice**] delivers a [**scripted narration**] [**in Polish**], likely from a [**recorded radio or podcast segment**], accompanied by [**subtle studio ambiance**] including [**microphone hiss**] and [**paper rustling**]. A [**secondary listener**] [**wearing headphones**] remains [**audibly inactive**], though [**faint page-turning sounds**] indicate [**preparatory material review**]. The spoken text references [**program materials available at Lechia.net**], suggesting a [**structured broadcast format**] with [**editorial oversight**]. Background contains [**minimal environmental noise**] consistent with a [**sound-treated recording space**].

Total flagged phrases: **17**

**Note**: The total number of flagged phrases is provided for reference. If you believe other words or phrases are important in the context of the verification, please consider them in your calculation as well.

YOUR TASK

- Listen to the audio and verify the highlighted elements.
- Assign one of the following error values to each phrase:

| Label | Criteria |
|-------|----------|
| Correct (0) | Directly verifiable from audio |
| Unverifiable (0.5) | Neither confirmed nor disproven, or things you are not sure |
| Hallucination (1) | Contradicts audio or invents content |

The final hallucination rate is calculated as follows:

$$\text{Hallucination Rate} = \left( \frac{\sum(\text{Error Values})}{\text{Total Content Units}} \right) \times 100\%$$

Based on the hallucination rate, assign a final score as follows:
**0-10%:** Score = 5 | **11-25%:** Score = 4 | **25-40%:** Score = 3 | **41-50%:** Score = 2 | **51-100%:** Score = 1

Figure 6: Instruction for Human Evaluation.

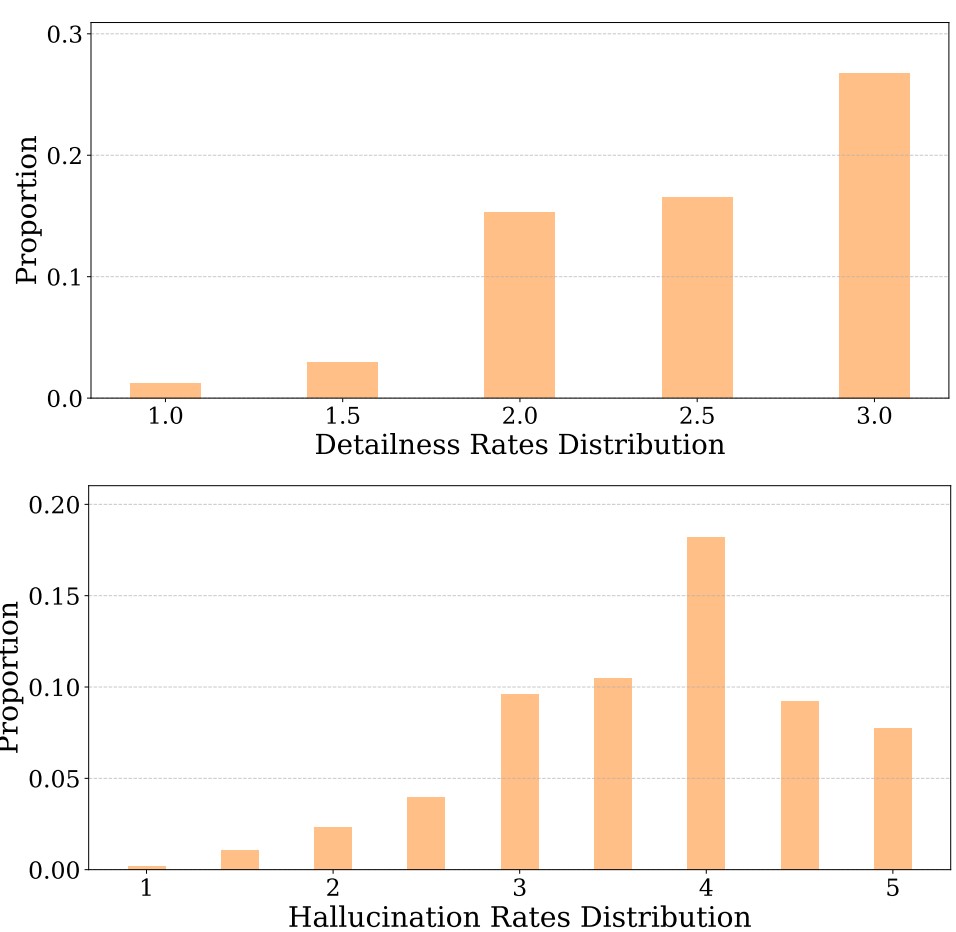

Figure 7: Detailness and Hallucination Rates Distribution of Human Rating

## C  PROMPT FOR MODELS

### C.1  MODAL CHECK PROMPT

The prompt we used to check if the modal information is used during the fusion is shown in Figure 8.

### C.2  VIDEO CAPTION PROMPT

The prompt we used for Qwen2.5-VL-72B to extract video caption is shown in Figure 9. We try to let the model describe sound-related object only, but found that it would introduce additional hallucinations. Thus, we prompt the model to describe visual content only, and let the integration model tackle the modality issue.

### C.3  AUDIO CAPTION PROMPT

The prompt we use for GAMA to extract audio caption is shown in Figure 10.

### C.4  OBJECT EXTRACTION PROMPT

The prompt for asking GPT-4o mini to obtain instruments, emotions, and music styles from audio is shown in Figure 11.

### C.5  MODAL INFORMATION CHECK PROMPT

---

**Shortened Prompt for Modal Integration Check**

```
"You analyze descriptions from audio.
'final_cap' is a comprehensive summary.
Identify source captions ('audio_caption', 'speech_caption',
'music_caption', 'video_caption') essential for 'final_cap'
using provided JSON data: {cap_str}.

Requirements:
1. List contributing caption types.
2. Return as string keys list.
3. Format: ['type1', 'type2']"
```

Figure 8: Concise prompt for modal info check

```
Prompt for video caption

Prompt:
Please provide a comprehensive video description focusing exclusively on observable visual elements,
including timestamps:
**1. Key Entities & Actions with Timestamps:**
- List main objects/subjects and their visible actions with approximate timestamps (MM:SS format)
- Describe:
* Object/subject movements and interactions
* Material properties (metal, wood, liquid)
* Timing of significant visual events

**2. Scene Description with Timeline:**
- Overall scene dynamics and visual interactions
- Notable visual events with timestamps:
* Object collisions or impacts
* Movement patterns
* Material changes
* Human/animal visible actions
- Environmental context (indoor/outdoor, spatial relationships)

**3. Overall Description with Chronological Flow:**
- Provide a comprehensive visual narrative of the video
- Include timestamps for key moments and transitions (MM:SS format)
- Focus on observable actions, and movements
- Use specific, action-oriented language
- Present events in chronological order with clear time markers

Guidelines:
- Describe only directly visible elements
- Focus on observable actions and movements
- Note material properties and physical interactions
- Include **timestamps** for all significant events
- Timestamp **should not** exceed the duration of the video
- Use precise descriptive language for visual elements
- Avoid assumptions about non-visible elements
- Maintain strict focus on visual information

Example:
Instead of "A car's engine roars as it accelerates"
Write "00:01 - A red sports car with chrome detailing accelerates down a paved road, tires creating
visible spray on wet asphalt"
"00:02 - The car's rear suspension compresses during acceleration, exhaust emitting visible vapor"
"00:03 - The car's engine roars as it accelerates"
```

Figure 9: Prompt for video caption.

```
An example prompt for audio caption generation

Describe the audio in detail, but there is not need for association or speculation.
```

Figure 10: An example prompt for audio caption generation

**An example prompt for extracting objects from audio**

I will give you a sentence. Please extract some information I need in a JSON format. Sentence: 'caption'
My requirement:
1. Extract instruments and return as a list
2. Extract emotions and return as a list
3. Extract music genres and return as a list
4. Extract scenes and return as a list
5. All words must be found in the sentence.
6. Return a JSON format without any other words.
7. Words must be extracted from the corresponding caption.

The return format should only be like this:

```
{
    "instrument": [],
    "emotion": [],
    "music genre": [],
    "scene": []
}
```

Figure 11: An example prompt for extracting objects from audio.

**Examples for audios with different clap scores.**

Here we show the severity of hallucinations in audio captions under different clap similarity intervals. The red - marked parts are the hallucinatory parts of the audio captions.

Figure 12: Examples for audios with different clap scores.

---

**An example prompt for multi-choice questions**

**Prompt:**
Please provide a comprehensive video description focusing exclusively on observable visual elements, including timestamps:
**1. Key Entities & Actions with Timestamps:**
- List main objects/subjects and their visible actions with approximate timestamps (MM:SS format)
- Describe:
* Object/subject movements and interactions
* Material properties (metal, wood, liquid)
* Timing of significant visual events

**2. Scene Description with Timeline:**
- Overall scene dynamics and visual interactions
- Notable visual events with timestamps:
* Object collisions or impacts
* Movement patterns
* Material changes
* Human/animal visible actions
- Environmental context (indoor/outdoor, spatial relationships)

**3. Overall Description with Chronological Flow:**
- Provide a comprehensive visual narrative of the video
- Include timestamps for key moments and transitions (MM:SS format)
- Focus on observable actions, and movements
- Use specific, action-oriented language
- Present events in chronological order with clear time markers

Guidelines:
- Describe only directly visible elements
- Focus on observable actions and movements
- Note material properties and physical interactions
- Include **timestamps** for all significant events
- Timestamp **should not** exceed the duration of the video
- Use precise descriptive language for visual elements
- Avoid assumptions about non-visible elements
- Maintain strict focus on visual information

Example:
Instead of "A car's engine roars as it accelerates"
Write "00:01 - A red sports car with chrome detailing accelerates down a paved road, tires creating visible spray on wet asphalt"
"00:02 - The car's rear suspension compresses during acceleration, exhaust emitting visible vapor"
"00:03 - The car's engine roars as it accelerates"

Figure 13: An example prompt for multi-choice questions.

---

**Prompt for integration**

**Prompt:**
Rigorous Multimodal Information Integration and Purely Audio Description Expert

**Core Task**
You are an expert specializing in audio information processing. Your goal is to: integrate and analyze textual descriptions from multiple modalities as input, perform cross-referencing and correction while strictly controlling cross-modal information interference, and ultimately generate a description that is **purely about the audio content**, accurate, detailed, and fluently written in English, annotating potential ambiguities **based solely on auditory perception**. **It is strictly prohibited to include any visual information, specific speech dialogue content, or ambiguity annotations based on audio-visual inconsistencies in the final output.**

**Input Information Sources (May contain errors, hallucinations, or be incomplete)**

- Audio Tags: A set of sound category tags annotated by humans, along with their corresponding quality estimations (confidence scores). Represents the most prominent human-perceived acoustic features in the audio. **These tags are highly reliable, especially those with high percentages**, but may not comprehensively cover all information in the audio. **The format is TagName(Percentage%). e.g., Speech(100%).** If empty, it indicates no human-annotated tag information is available.

- Audio Description: A textual description of the audio content (may include sound events, ambient sounds, music, vocal characteristics, etc.). This is an **important basis** for describing audio facts and needs to be cross-validated with tags and music descriptions.

- Speech Content: The textual result from Automatic Speech Recognition (ASR). **This information is used only to confirm the presence of human voice, determine general vocal characteristics (e.g., speech vs. non-linguistic sounds, presence of distinct emotions [non-content related]), and assist in inferring possible scenarios or event backgrounds. Its specific textual content (including paraphrasing or summarization) must never appear in the final output.** If empty, it indicates no distinct human voice, or other non-linguistic vocalizations (e.g., gasping, crying, background babble).

- Music Description: A description of musical elements (features, instruments, rhythm, etc.) and other sound scenes. **Music-related features herein are highly reliable. If empty, it indicates no distinct music.** Other non-music descriptions (e.g., environment, human voice) have lower priority and primarily depend on "Audio Tags", "Audio Description", and "Speech Content" for judgment.

- Video Description: A textual description of the video frames. **Used only under specific conditions** (see "Active Correction" in Processing Steps, step 2) **to actively assist in identifying auditorily ambiguous sound sources**, and **to identify inconsistencies with auditory information (this inconsistency is only an internal decision-making flag for the model, not used to generate the output ambiguity list)**. **Never** speculate or describe the source, location, or on-screen actions of sounds based on video information itself. If empty, it indicates a lack of visual auxiliary information.

**Processing Steps**
Please strictly follow the steps below:

1. **Multimodal Information Parsing**:
   - Separately interpret each input description to extract core sound events, sound source characteristics, environmental ambiance, and musical elements.
   - Specifically parse "Audio Tags" to extract tag names and their confidence scores.
   - **Special Note**: From "Speech Content" (ASR results), primarily determine **if human voice is present** and its **non-content features**. In conjunction with its textual content (**used only for auxiliary understanding**), **assist in inferring** possible environments, emotional tones, or types of acoustic events, but **never judge** speaker gender, age, or other personal characteristics based on ASR content, and **never quote, paraphrase, or summarize** the specific textual content.

Figure 14: Prompt for integration.

---

**Prompt for integration Cont.**

2. **Auditory Fact Determination and Cross-modal Correction**:

- **Initial Determination of Auditory Facts**: First, based on `"Audio Tags"` (**especially high-confidence tags, which have the highest priority for determining the types of sounds included in the tags**), `"Audio Description"`, `"Music Description"` (especially the music part), and `"Speech Content"` (presence of human voice and inferred characteristics), preliminarily determine auditorily perceived sound events, sound sources, ambient sounds, and music features. Identify and attempt to correct contradictions within these audio information sources (tags, audio description, music description, ASR inference), with the priority rule: **High-confidence `"Audio Tags"` > Music part of `"Music Description"` ≈ `"Audio Description"` > `"Speech Content"` (presence of human voice) > Low-confidence `"Audio Tags"` > Non-music part of `"Music Description"`**.

- **Cross-modal Validation and (Conditional) Active Correction (for video information)**: After the initial determination of auditory facts, introduce `"Video Description"` for cross-validation. Its role is:
  - **Active Correction (when audio information is ambiguous and video provides clear evidence):** If the initially determined auditory fact (based on audio information sources) describes a **general sound type that could have multiple auditory interpretations** (e.g., a rumbling sound, a clicking sound, a rustling sound), **and** the `"Video Description"` clearly shows an object or event that is **highly relevant to this general sound type and is a plausible sound source** (e.g., the video clearly shows an airplane making a rumbling sound, or a person clicking a mouse making a clicking sound, or clothes/fabric in motion making a rustling sound), then **the information provided by the video should be adopted to more precisely identify the general sound as a specific source or type** (correcting rumbling to airplane sound, clicking to mouse click, rustling to fabric rustle). **Note: If a high-confidence tag in `"Audio Tags"` already clearly indicates the specific sound type, then this sound is no longer considered a 'general sound type with multiple auditory interpretations,' and this active correction step no longer applies to this sound.** Under these limited and clear conditions, video information is used to **enhance** the understanding of audio facts, making the description more precise.
  - **Identifying Inconsistencies or Lack of Corroboration (when video cannot clearly corroborate or conflicts):**
    * If the sound event described by the initially determined auditory facts **does not have a clearly corresponding visual sound source** in the `"Video Description"`, or if the visual information **is inconsistent with or contradicts the perceived location or state of the sound source**, then **video information must never be used to negate or modify known auditory facts**. In such cases, the model should **internally flag** the presence of an audio-visual inconsistency or lack of visual corroboration. **This flag is only used in subsequent steps to adopt conservative wording when generating the final audio description and must never directly generate an ambiguity entry for output.**
    * **It is strictly prohibited to speculate, describe, or alter judgments about the sound event itself based on video information that cannot corroborate the audio (e.g., hearing a rumbling sound, the video shows the sky, but one cannot speculate it's an airplane sound unless the video explicitly shows an airplane).**

- **Determine Corrected Auditory Facts**: Based on the results of the above multimodal cross-validation, determine the final auditory facts. The priority rule is listed above. **When cross-modal information conflicts, audio information sources conflict internally, or there is high uncertainty (especially a lack of high-confidence tags or clear video corroboration for audio) making it difficult to determine auditory facts, the determined facts should reflect extreme conservatism, preferring to omit uncertain information rather than speculating based on non-auditory information.** The model should internally retain a flag for the uncertain origin of audio information (e.g., whether it's due to a lack of high-confidence tags, lack of support from audio description, or lack of video corroboration), to generate appropriately conservative descriptions in step 5.

- **Emotion Inference and Correction**: If the emotion of a sound event (e.g., human voice, whose emotion can be inferred with ASR content assistance) conflicts with the emotion of background music, a comprehensive judgment must be made to provide the most likely primary emotional tone, but this is still based on auditory and ASR-assisted inference, without introducing visual information.

3. **Purely Auditory Ambiguity Reasoning and Annotation**:

- **Focus Solely on Pure Audition**: Based on the **determined auditory facts** (which have considered tags and correction results), sound characteristics, common possibilities of auditory confusion, and potential auditory understanding biases in the original audio description, infer potential auditory understanding ambiguities that can be **perceived or reasonably inferred solely through hearing**.

Figure 15: Prompt for integration Cont.

Prompt for integration Cont.

4. • **Sources of Ambiguity**:
  – **Auditory Similarity or Vagueness of the Sound Itself**: Some sounds may be auditorily similar to others and easily confused (e.g., vehicle sound vs. airplane sound, typing sound vs. light tapping sound). The sound's own quality, distance, or reverberation can also lead to vagueness or difficulty in determining the source.
  – **Polysemy of Auditory Association**: A sound event may reasonably correspond auditorily to multiple different sound sources or situations (e.g., a "bang" can have multiple causes, footsteps might come from multiple people).
  – **Potential Purely Auditory Biases in the Original Audio Description**: If, after multimodal correction, the original `"Audio Description"` is found to have incorrect or imprecise judgments about sound events or sources (and this error/imprecision is not caused by audio-visual inconsistency but by potential misinterpretations of audition itself), one should infer what common purely auditory misinterpretations the original description might have been based on.
  • **Strictly Exclude Non-Auditory Information as a Source of Ambiguity**: Ambiguity annotation must **only** revolve around pure auditory perception and the associations arising therefrom. **It is absolutely not allowed** to use audio-visual synchronization, the way sound sources are presented on screen, or any visual content as the source or descriptive content of an ambiguity.

5. **Information Reliability Assessment and Final Output Decision**:
  • **In this step, based on the analysis and correction results from steps 1-3, comprehensively assess the reliability and completeness of the determined auditory facts. In particular, consider whether high-confidence audio tags support key sound events.**
  • **If it is judged that the determined auditory facts are extremely scarce, various audio information sources (tags, audio description, music description, ASR inference) severely conflict and auditory facts cannot be reliably reconstructed, or even if tags exist but their confidence is generally very low and contradicts other information, the model directly outputs the unique specific string `UNCERTAIN_AUDIO_INFORMATION_DETECTED`.**
  • **Otherwise (if the determined auditory facts are sufficiently reliable and complete), proceed to the next step (generating JSON).**

6. **Generate Final Pure Audio Description (Audio Caption)**:
  • **Execute this step only after passing the reliability assessment in step 4.**
  • **Pure Audio Focus**: Generate a fluent, accurate, detailed, and concise English audio description. **Describe only what can be heard and its purely auditory characteristics** (e.g., sound source type [prioritizing those confirmed by high-confidence tags or clearly identified through active video correction], nature of sound events, type of ambient sound, music features, non-content features of human voice, spatial sense, loudness, timbre, duration, rhythm, etc.).
  • **Integration and Augmentation**: Integrate all valid auditory facts determined after multimodal correction (including those from audio tags, audio description, music description, ASR inference, and sound source types actively corrected via video). Supplement necessary auditory details of the scene (e.g., indoor/outdoor inferred from ambient sounds). **If the model has internally flagged uncertainty in the audio information** (e.g., lack of high-confidence audio tags supporting key sound events, original audio description being auditorily vague and lacking clear video corroboration, or internal conflicts within audio information sources), **the final description must reflect this uncertainty, but through cautious wording to describe the perceived sound itself, rather than directly stating the uncertainty or vagueness.** Use phrases like "sounds like," "appears to be," "potentially," "suggests," "a sound resembling X is heard" to express identification of less certain sound sources or events. **Crucially, avoid sentences that explicitly state an inability to determine something or that something is ambiguous (e.g., do not say "the source cannot be determined" or "it is ambiguous whether X is present"). Instead, directly omit highly uncertain details or use cautious wording for what *might* be perceived.**
  • **Objective and Accurate**: Base inferences on determined auditory facts, avoiding subjective speculation and over-extension. The description content must be supported by input text. Prohibit the introduction of irrelevant "new information," unless it is reliable auditory inference based on multiple audio information sources (e.g., inferring the scene from ambient sounds). Ensure the description integrates facts confirmed by high-confidence tags, but **never mention the confidence percentages themselves.**
  • **Cultural/Emotional Cues**: If the sound contains clear cultural symbols or strong emotions, these can be briefly cued, but must be based on input audio evidence (e.g., emotion in human voice inferred from ASR, or emotion reflected by music features).
  • **Final Check**: Ensure this description **absolutely contains no visual elements** (objects, colors, actions, visual scenes, etc.). Even if the sound source type has been determined through high-confidence tags or active video correction, **never describe the visual location, visual form, or specific on-screen behavior of that sound source**. **Absolutely prohibit** the output of any specific speech text content (quotation, paraphrase, summary).

Figure 16: Prompt for integration Cont.

---

**Prompt for integration Cont.**

**Output Format Requirements**

**For most cases (i.e., when passing the reliability assessment in step 4), please strictly generate structured English output in the following JSON format (without any other explanations). However, in the special case of "scarce/unverifiable information" defined in step 4 of the processing flow, the model should directly output the predefined string** `UNCERTAIN_AUDIO_INFORMATION_DETECTED` **instead of JSON.**

```
{
    "Potential ambiguities": [ // List potentia ambiguities based purely on
    auditory perception (English sentences). Does not include ambiguities
    requiring visual information
    to understand, nor ambiguities based on audio-visual inconsistencies.
        "Ambiguity description 1 based solely on auditory perception.",
        "Ambiguity description 2 based solely on auditory perception.",
        ...
    ],
    "Audio caption": "Final audio description focusing solely on audible elements
    and their auditory characteristics, detailed and fluent English. Use
    conservative language when audio facts are uncertain based on internal
    assessment."
    // Final pure audio description (concise and clear English sentence)
}
```

**Key Considerations**:

- Output Language: **English**.

- Ignore Empty Inputs: If a modal description is empty, ignore that information source.

- **Strictly Prohibited**: Including any visual information (objects, colors, actions, visual scenes, visual location/form/on-screen behavior of sound sources, audio-visual synchronization, etc.) in the final output (including `Audio caption` and `Potential ambiguities`). Even when dealing with audio-visual inconsistencies or unknown sound sources, never speculate, describe, or mention any visual content in the output.

- **Strictly Prohibited**: Including any specific speech text content (quotation, paraphrase, summary, etc.) in the final output. Speech information is only used to infer the presence of human voice, non-content features, and to assist in understanding the scene ambiance.

- Maintain Objectivity: Base inferences on determined auditory facts, avoiding subjective speculation and over-extension. Information not supported by the input or derived through reliable auditory inference must not appear in the output.

- When the model internally flags audio information as uncertain (even if `UNCERTAIN_AUDIO_INFORMATION_DETECTED` is not triggered), the final `Audio caption` must strictly use cautious wording to describe the sound itself, focusing on auditory perception. **It is strictly prohibited to directly state uncertainty or ambiguity; only provide confirmed acoustic facts and do not mention uncertain content in the output.**

- **High-confidence audio tags are the highest priority source for determining sound type facts, but specific confidence values are not allowed in the output.**

Figure 17: Prompt for integration Cont.

## C.6 CASE STUDY

Table 7 presents representative FusionAudio captions, annotated with their information sources. These examples illustrate FusionAudio's ability to synthesize and reason across modalities, generating descriptions that go beyond mere aggregation to provide holistic, context-rich interpretations.

Table 7: Example of FusionAudio caption generation with annotated information sources.

| | |
|---|---|
| **Audio Caption** | The audio is characterized by the sound of brass instruments, possibly trumpets, playing music. |
| **Speech Caption** | (empty) |
| **Music Caption** | This is an instrumental marching band piece. The brass section plays the melody, while bagpipes provide a counter-melody. The mood is optimistic and cheerful. This piece may be used as background music for a national patriotic event. |
| **Video Caption** | Chronological overall description:
- **00:00**: The video opens with close-ups of musicians, highlighting their focused expressions and the shiny brass instruments. The musicians are seated in a row, each holding a saxophone or trombone, preparing to play.
- **00:01**: As the conductor steps forward facing the musicians, a blue-and-white striped shirt appears in the frame. The conductor's gestures begin to guide the musicians as the performance starts.
- **00:02 - 00:10**: The camera remains relatively still, capturing the musicians playing their instruments. The conductor's hand movements synchronize with the music, guiding the ensemble. The musicians' fingers move rhythmically on the keys and valves, producing harmonious sounds. The conductor continues to lead, ensuring cohesion.
- **00:07 - 00:10**: The camera pans slightly right, revealing more trombone players at the far right. The additional musicians add depth to the ensemble, enhancing the overall sound and visual impact. The video ends with the musicians continuing to play under the conductor's direction. |
| **Reasoning Content** | **The audio caption mentions "trumpets," but the music and video captions indicate a brass ensemble with saxophones, trombones, and bagpipes. The music caption is more reliable for instrument identification, so the final description should prioritize saxophones, trombones, and bagpipes.**

**There is no speech content, which matches the empty speech caption.**

**The mood is cheerful and ceremonial, suitable for a patriotic event, as inferred from the music caption.**

**Potential ambiguities include confusion about the specific types of brass instruments and the presence of bagpipes, as well as uncertainty about whether the performance is indoors or outdoors.**

... (Further details omitted for brevity) |
| **Final Caption** | A lively instrumental piece performed by a brass ensemble, with saxophones and trombones carrying the main melody and bagpipes providing a contrasting counter-melody. The music maintains the rhythmic precision characteristic of a formally conducted ensemble, featuring the harmonious and dynamic qualities typical of patriotic or ceremonial performances. No prominent vocal content. |

## C.7 SAMPLES OF DIFFERENT CLAP SCORE

Table 8: The demonstration of the hallucination which is marked in red of audio captions within different clap similarity intervals

| Clap Similarity Intervals | Audio ID | Caption |
| --- | --- | --- |
| 0.0-0.1 | -wyJ2cab4ic | A speech with strong tonal urgency is delivered, accompanied by persistent breathing sounds and faint intermittent background activity suggesting an indoor environment. The speaker's vocal cadence appears strained, potentially reflecting either passionate delivery or underlying emotional tension. |
| 0.1-0.2 | -4t1LMiiHp4 | A clear male speech is delivered with a strong vocal presence, accompanied by dynamic acoustic drums, a groovy bassline, and intermittent tambourine shakes in the background. Sporadic applause and crowd cheering weave throughout the speech, creating an energetic and engaged atmosphere. The musical elements maintain a steady rhythmic foundation while the vocal delivery appears deliberate and focused. |
| 0.2-0.3 | 04Q_WeM7VIU | Continuous music with a groovy bass line, percussive drum patterns, keyboard harmonies, and synth brass melodies is heard in a lively setting. Intermittent male speech occurs in an upbeat tone, overlapping with the music's rhythmic elements. The recording exhibits mono audio and background noise, suggesting a live performance environment with frequent equipment adjustments and energetic vocal exchanges. |
| 0.3-0.4 | -CCsZneHL6s | A solo violin performs a slow, emotive melody with a smooth bowing technique, accompanied by steady rhythmic percussive sounds suggesting a handpan or similar instrument. The performance takes place in an indoor environment with subtle background reverberation, indicative of a studio or concert space. The audio quality is slightly degraded, but the interplay between the sustained violin tones and precise percussive elements creates a harmonious, intimate atmosphere. |
| 0.4-0.5 | -EKjvd8q_A0 | The audio features a lively and energetic performance with rhythmic maracas, congas, and an accordion, accompanied by a saxophone adding depth. The upbeat tempo and festive soundscapes suggest a cultural celebration or live musical event. |
| 0.5-0.6 | 00Twebqicmo | The audio is dominated by powerful car engine revving and acceleration sounds, accompanied by continuous background music. The combination of loud mechanical noises and energetic musical accompaniment creates a high-intensity atmosphere characteristic of an automotive event. Intermittent engine echoes suggest open-air acoustics typical of a racetrack or exhibition setting. |

Table 8 presents the hallucination situations of FusionAudio captions within different CLAP similarity intervals.

## C.8 SITUATIONS WHERE MULTIMODAL CONTEXTUAL CUES WORK

Our multimodal approach is designed to excel in challenging audio understanding scenarios (Table 9), such as interpreting audio in adverse conditions, achieving high-level semantic understanding (e.g.,

Table 9: Key use-case scenarios where integrating multimodal contextual cues can significantly improve audio captioning. Challenges are listed per sub-scenario. Representative datasets and samples are detailed in Appendix C.9.

| Scenario | Sub-Scenario | Key Challenges |
|---|---|---|
| Adverse Acoustic Conditions | Scene Recognition in Complex Soundscapes | High inherent acoustic complexity; Interwoven multi-source information; Background noise |
| | Acoustically Degraded Conditions | Recording device limitations; Synthetic Artificial noise interference |
| High-Level Semantic Understanding | Music Understanding | Musical Genre Analysis; Emotional Expression; Artistic Intent; Aural Narratives |
| | Sound Understanding | Sound Implied Information; Attributes Inference |
| Fine-grained Information Recognition | Acoustic Entity Recognition | Subtle acoustic cue discernment |

nuanced music interpretation), and enabling fine-grained acoustic entity recognition. Addressing these scenarios highlights the benefits of comprehensive multimodal integration.

## C.9  SAMPLES OF DIFFERENT SUB-SCENARIO

Table 10: Dataset and examples corresponding to each sub-scenario, where cls is the classification task

| Sub-Scenario | Datasets(quantity) | Examples |
|---|---|---|
| Scene Recognition in Complex Soundscapes | AIR-Bench:
    Acoustic scene cls(2,000)
UrbanSound8K(8,732) | *Identifying child playing scene*
*Identifying kitchen scene* |
| Acoustically Degraded Conditions | TAU Urban Sound-Mobile(5,265)
FSDnoisy18K(947) | *Identifying street pedestrian sound*
*Identifying metro station scene* |
| Music Understanding | AIR-Bench:
    Genre cls(2,000)
MusicAQA(814)
Mood detection(2,000)
Chat-Music(500) | *Identifying music genre*
*Character portrayed by the tune*
*Trumpet&accordion's role in texture* |
| Sound Understanding | AIR-Bench:
    SoundAQA(2,000)
    Chat-Sound(500)
AudioBench:
    Audio-Scene QA(9,131) | *Location of dripping water*
*Possible actions with the liquid*
*Indications of a busy road* |
| Acoustic Entity Recognition | AIR-bench:
    Vocal sound cls(1,000)
    Music instruments cls(2,000)
ESC-50(2,000)
FSD50K(10,231) | *Instrument recognition*
*Acoustic event/ontology recognition*
*Acoustic scene type recognition* |

Table 10 shows the example dataset for each sub-scenario and corresponding example samples.

# D MORE ON DATASET STATISTICS

## D.1 EMBEDDING SPACE QUANTITAVE ANALYSIS

Table 11 presents a comprehensive comparison of inter- and intra-category embedding distances across different datasets. The analysis focuses on three key audio categories: Music (M), Vehicle (V), and Speech (S). Our proposed FusionAudio dataset demonstrates superior performance across all metrics. For inter-category distances, where higher values indicate better category separation, FusionAudio achieves significantly larger distances between different audio types (M-V: 0.7230, M-S: 0.5369, V-S: 0.5943) compared to competing datasets. This indicates that our dataset enables models to learn more discriminative representations that effectively distinguish between different audio categories. Simultaneously, FusionAudio exhibits smaller intra-category distances (Music: 0.8084, Vehicle: 0.7406, Speech: 0.8204), reflecting greater consistency within each category. The substantial improvement in both metrics—maximizing inter-category separation while minimizing intra-category variation—confirms that FusionAudio produces more cohesive and well-structured embedding spaces. This balance is crucial for downstream tasks such as audio classification, retrieval, and generation, as it facilitates more accurate identification and characterization of audio content while maintaining the nuanced variations within categories.

Table 11: Inter- (M-V, M-S, V-S) and Intra- (M, V, S) category embedding distances. Best inter-distances (higher) and intra-distances (lower) are bolded.

| Dataset/Method | Inter-category distance ↑ | | | Intra-category distance ↓ | | |
|---|---|---|---|---|---|---|
| | M – V | M – S | V – S | Music | Vehicle | Speech |
| FusionAudio | **0.7230** | **0.5369** | **0.5943** | **0.8084** | **0.7406** | **0.8204** |
| ASC | 0.5685 | 0.4137 | 0.4523 | 0.8638 | 0.8216 | 0.8724 |
| Auto-ACD | 0.5685 | 0.4137 | 0.4523 | 0.8645 | 0.8402 | 0.8915 |
| Sound-VECaps | 0.5232 | 0.3770 | 0.4664 | 0.8578 | 0.7798 | 0.8920 |

# E    ABLATION STUDY

## E.1    ABLATION STUDY ON FILTER MODULE IN AUDIO UNDERSTANDING TASK

As Table 12 shown, our filtered dataset outperforms the unfiltered version in 14 out of 15 individual benchmarks and demonstrates superior average performance across all three main evaluation categories. This conclusively proves that our filtering method is a principled process that demonstrably improves downstream task performance by removing text-audio pairs with significant content mismatch.

Table 12: Ablation result on Filter Module in Audio Understanding task

| Dataset | ASC (Acc.) | TAU (mAP) | FSDn (mAP) | $US_{8k}$ (mAP) | Gnr (Acc.) | $M_{AQA}$ (Acc.) | Mood (Acc.) | $C_M$ (M.J) | $S_{AQA}$ (Acc.) | $C_S$ (M.J) | $AB_S$ (M.J) | Voc (Acc.) | Ins (Acc.) | ESC (mAP) | FSD (mAP) |
|---|---|---|---|---|---|---|---|---|---|---|---|---|---|---|---|
| Unfiltered | 52.62 | 21.61 | 85.83 | 61.13 | 59.20 | 55.28 | 33.40 | 57.37 | 56.40 | **62.32** | 62.80 | 65.50 | 70.50 | 66.48 | 42.90 |
| Filtered | **59.68** | **25.12** | **88.20** | **63.99** | **64.20** | **59.95** | **38.30** | **57.94** | **58.35** | 62.28 | **63.96** | **71.00** | **73.85** | **71.27** | **47.37** |

# F    HUMAN PREFERENCE ARENA EVALUATION FOR CAPTION QUALITY

## F.1    AREANA DETAILS AND RESULTS

While benchmarks in main text confirm our model's SOTA performance, we posit that they do not fully capture the qualitative nuances of our primary contribution: generating fine-grained, high-fidelity audio captions. Automated metrics are limited by the "one-to-many" problem (penalizing diverse, correct answers) and can paradoxically punish models for providing more correct detail than is present in a reference.To overcome these limitations, we conduct a head-to-head human evaluation using a pairwise comparison (Arena) methodology, assessing Detail and Accuracy (low hallucination). This directly tests the trade-offs between descriptive richness and factual correctness. Table 13 shows the result of Arena.It proves that on Accuracy (i.e., low hallucination), our model ranks #1, surpassing all other models. While Gemini is very detailed, this comes at the cost of a significantly higher rate of hallucination, where it invents facts not present in the audio. Our method achieves a superior balance, providing substantial detail while maintaining the highest factual fidelity.

Table 13: Human Evaluation Results of Audio Caption Models (Based on Arena Methodology)

| Model | Evaluations | Detail Rank | Detail ELO | Detail (W/L/T) | Accuracy Rank | Accuracy ELO | Accuracy (W/L/T) |
|---|---|---|---|---|---|---|---|
| gemini-2.5-pro | 81 | #1 | 1284 | 61 / 10 / 10 | #2 | 1080 | 39 / 20 / 22 |
| fusionaudio-high-25k | 84 | #2 | 1103 | 52 / 22 / 10 | **#1** | **1118** | **51 / 14 / 19** |
| gpt-4o-audio-preview | 82 | #3 | 1034 | 37 / 36 / 9 | #3 | 1035 | 34 / 28 / 20 |
| qwen2.5-omni-7b | 79 | #4 | 793 | 9 / 62 / 8 | #5 | 847 | 9 / 54 / 16 |
| gama-it | 50 | #5 | 785 | 8 / 37 / 5 | #4 | 920 | 10 / 27 / 13 |

