# OpenReview forum: "Towards Fine-grained Audio Captioning with Multimodal Contextual Fusion"
_ICLR.cc/2026/Conference — ICLR 2026 Conference Withdrawn Submission_

### Official Review · Reviewer_fb6q · 2025-10-26

**Soundness:** 2
**Presentation:** 2
**Contribution:** 3
**Rating:** 2
**Confidence:** 5

**Summary:**

This paper introduces a multimodal framework for generating fine-grained captions for audio and video content. The proposed system operates in two stages:

(1) multiple specialized models (such as ASR models for speech, music captioning models for music, and video captioning models for video) are employed to extract modality-specific features and descriptions;

(2) an LLM-based module integrates these heterogeneous captions into a coherent and contextually consistent description.
The authors demonstrate that their approach can improve downstream performance in audio retrieval and audio understanding tasks, showing the potential of multi-source caption fusion.

**Strengths:**

•  The framework effectively combines different audio and video analysis models, enabling a more comprehensive and fine-grained understanding of multimodal content.

•  The proposed dataset and experiments show consistent improvements in retrieval and understanding benchmarks, suggesting that the generated captions contain richer semantic information.

**Weaknesses:**

•  The overall framework is structurally similar to Sound-VEcaps, with the primary difference being the inclusion of more domain-specific feature extraction models such as ASR model for speech and Music model for music caption.

•  The paper lacks comparative analysis on audio generation tasks, which could better demonstrate the generalization of the learned representations.

•  The quality filtering strategy used for caption selection is not well-detailed, leaving uncertainty about its effectiveness and criteria.

•  While quantitative results are promising, the paper would benefit from qualitative comparisons or demo examples showcasing differences between LLM-based captions and traditional model-generated captions.

**Questions:**

•  Have you conducted any experiments on audio generation tasks ?

•  Could you provide more details about the quality filter module, including how filtering thresholds are determined and what metrics are used?

•  Would it be possible to include demo comparisons between different LLM-based captions to better illustrate the qualitative improvements?

•  Is there any ablation study showing that these vision-based information affacts the overall performance of different downstream tasks?

---

> ### Author Response · Authors · 2025-12-04
>
> We sincerely thank the reviewer for the insightful questions. We will make the following comments to address the review's concern.
> ### **1. The reviewer argues that our work lacks novelty, viewing it as a straightforward integration of existing models rather than a methodologically innovative contribution.**
> * **Response on Limited Novelty:**
> We respectfully argue that our main contribution lies in a methodological advance: a principled pipeline and the large-scale dataset constructed through it, both of which directly address a critical bottleneck in audio-language research—the lack of a low-hallucination, verifiable automatic data pipeline and large-scale, high-quality audio–text data. Rather than a “straightforward integration,” our system is a carefully designed architecture that aims to mimic sophisticated human perception. In particular, our two-stage caption generation pipeline is novel in how it provides diverse, structured context to the LLM for detailed synthesis, moving beyond simple feature concatenation toward complex, guided reasoning.
>
> * **Methodological Depth:**
> Our fusion strategy extends beyond which modalities to combine, to how they are synthesized. We invested significant effort in each modalities and how to effectively integrate this information module. This provides valuable and non-trivial practical insights for the large-scale scenario of using LLMs for automatically constructing data. Furthermore, our choice of a powerful LLM was deliberate. We established the necessity of using a strong reasoning model, whose advanced Chain-of-Thought (CoT) capabilities are essential for handling the complex cross-modal inference and conflict resolution required to produce higher-fidelity, less-hallucination and verifiable-process captions, which is something other related works couldn't achieve. This demonstrates the task's complexity and the sophistication of our solution.
>
> * **Novelty Demonstrated by High Data Efficiency and Impact:**
> The ultimate validation of our method's novelty is the quality and utility of the data it produces. Our models achieve state-of-the-art performance while using significantly fewer unique audio clips (Sec 5.1.2). This demonstrates that our method successfully extracts more high-quality, fine-grained information per sample—a direct result of our novel generation and synthesis pipeline and a clear step beyond prior art.
>
> In summary, while we use existing components, our novelty lies in the architecture that combines them, the sophisticated reasoning process we engineered for the LLM, and the resulting high-quality, high-efficiency data that fills a recognized gap in the field.
>
> ### **2. The experiment of downstream task: audio generation**
> Thank you for pointing out this interesting direction. We fully agree that conditioning audio generation models on our learned representations is a valuable way to further probe their generalization ability.
>
> However, our current work focus on scalable generation of fine-grained captions, and training LALMs with improved understanding and instruction-following capabilities. Evaluating audio generation would require integrating our data and pipeline with specific generative architectures and running large-scale training/inference, which is computationally expensive and requires careful design of additional baselines.
>
> Due to these resource and time constraints, we were not able to conduct a thorough and well-controlled set of audio generation experiments within the scope of this submission. Instead, we evaluate generalization of the learned representations across multiple audio understanding tasks (classification, retrieval, QA, etc.) and on external datasets, which we believe already demonstrate strong cross-task and cross-domain generalization.
>
> Systematic evaluation on audio generation tasks is a promising extension of FusionAudio, and we consider it important future work.
>
> ### **3. Quality Filtering Strategy and Evaluation Criteria**
> We use the CLAP model to generate embeddings for audio and captions respectively, and calculate the cosine similarity between them.Data with cosine similarity below a certain threshold will be filtered out.It is also a filtering method adopted in similar works(AudioSetCaps: An Enriched Audio-Caption Dataset using Automated Generation Pipeline with Large Audio and Language Models；Audio Dialogues:Dialogues dataset for audio and music understanding；Synthio: Augmenting Small-Scale Audio Classification Datasets with Synthetic Data).

---

> ### Author Response · Authors · 2025-12-04
>
> ### **4. Effectiveness**
> **Threshold Selection Process：** We sampled 1,300 data samples, formulated rules to determine data eligibility, and then conducted manual annotation. The selection of the 0.08 CLAP similarity threshold was not arbitrary or chosen to match the hallucination rate. Instead, it was empirically determined as the value that maximized the F-score when evaluated against human preference judgments for caption quality. At this threshold, the alignment rate between the CLAP-based method and manual filtering reaches 88.3%.
>
> **Filtering Ratio：** According to statistics, approximately 7% of the data in the human-annotated dataset was filtered out. After applying the CLAP-based filtering scheme to the entire dataset, the filtering ratio was around 8.3%, which is still close to the filtering ratio from human annotation.
>
>
> ### **5. Lack of comparisons among different LLM-based methods**
> To better demonstrate the advantages of our work, we present several comparative cases with other related studies—all of which are generated based on large language models (LLMs).
>
> #### **case1**
> "AudioSetCaps": "This audio clip features a distinct, rich bell sound, creating an evocative and resonant sonic experience.",
>
> "Auto-ACD": "Soft ambient music plays as a clock ticks, creating a serene atmosphere in an indoor setting.",
>
> "Sound-VECaps": "Music is playing softly in the background while a person works on their laptop in a peaceful home office, surrounded by plants and natural light, with occasional snorts indicating a moment of amusement or surprise.",
>
> "FusionAudio": "The audio features a continuous ringing sound resembling a bell layered over a hypnotic, whining pad guitar in a monaural recording. Intermittent bird-like chirps emerge alongside shimmering hi-hat rhythms. As the piece progresses, the music transitions into a noisier, more mysterious tone while retaining subtle peaceful elements."
>
> **Analysis:**
> * **Richness of sound types:** In addition to identifying the bell sound recognized by most captions, our method further supplements bird chirps and crisp hi-hat sounds.
> * **Granularity of specific sound types:** Our captions clearly specify the instrument type as a whining pad guitar, whereas other captions merely indicate the presence of music.
>
> **model output:**
> "gpt-4o-audio-preview": "The audio contains a recurring, rhythmic tick-tock sound, reminiscent of a clock or metronome. The pattern is steady and consistent, creating a sense of timing and pace. The tone is relatively soft yet distinct, with no sharp or harsh elements. The sound is clear, without background noise. The volume is moderate, neither too loud nor too quiet, making each individual tick and tock easily distinguishable. There's also a slight mechanical or metallic resonance to the sounds, adding to the impression of a clockwork mechanism. There are no other prominent instruments, tones, or effects besides the repeated ticking sound. The overall acoustic atmosphere is calm and measured, with no abrupt changes or silences."
>
> "gemini-2.5-pro": "The audio begins with a high-pitched, sustained, and shimmering synthesized pad. It has an ethereal and slightly dissonant quality, with significant reverb and delay creating a spacious, ambient atmosphere. Layered underneath is a soft, continuous rattling or shaking sound, like a maraca or shaker. The volume is moderate. The audio then abruptly cuts to a loud, low-frequency, harsh, and jarring synthesized buzzing tone. This final sound is dry and direct, with no reverb."

---

> ### Author Response · Authors · 2025-12-04
>
> #### **case2:**
> "AudioSetCaps": "Male child makes a sound as he interacts with a scurrying mouse, creating an engaging and lively audio clip.",
>
> "Auto-ACD": "A woman speaks in a small room, with a probability of 0.85, according to the audio tags.",
>
> "Sound-VECap": "A child is speaking and then holds a toy in front of their face, followed by partially obscuring their face with a black object, all while sitting in a dimly lit room with a white door and vent.",
>
> "FusionAudio": "The audio primarily features a child's clear, continuous speech with intermittent soft mechanical noises resembling a small device or toy in the room. The child's voice remains dominant throughout, while the periodic mechanical sounds appear to accompany an interactive activity involving controlled movements."
>
> **Analysis：**
> The core content of the audio consists of a young boy's speech and mechanical white noise, and only FusionAudio can describe them accurately.
>
> **model output:**
> "gpt-4o-audio-preview": "The audio features a clear and softly spoken voice, with minimal background noise. The tone is calm and deliberate, suggesting a careful explanation or description. The volume is steady and moderate, without any sudden spikes or drops. There are no musical instruments or mechanical noises, just the pure sound of a single voice speaking at a consistent pace. No effects like echoes or reverberation are noticeable, and there seems to be a focused, almost precise delivery in the sound."
>
> "gemini-2.5-pro": "The recording begins with a low-fidelity, slightly muffled sound of speech in a small, untreated acoustic space, accompanied by faint rustling and handling noises in the background. At 00:09, this is abruptly interrupted by a loud, harsh, and sustained electronic buzzer tone that continues until the end of the recording."
>
> #### **case3:**
> "AudioSetCaps": "A male speaker is heard in the audio, accompanied by crowd cheers and water splashing sounds. The speaker's emotion cannot be discerned from the audio alone.",
>
> "Auto-ACD": "An adult male delivers a speech while a crowd enthusiastically cheers and applauds in an auditorium.",
>
> "Sound-VECaps": "A man yelling followed by a man screaming while wind blows into a microphone, as a zip line runs across a picturesque coastal landscape with a beach, town, and rolling hills, on a sunny day.",
>
> "FusionAudio": "A dynamic auditory scene features clear human vocalizations including shouts, screams, and laughter. These are accompanied by consistent rhythmic sounds that maintain a steady tempo throughout, creating contrast between the active vocal elements and the repetitive background noise."
>
> **Analysis：**
> The core of the audio lies in humans' joyful shouts and the sound of the zip line, and only FusionAudio can describe them accurately.
>
>
> **model output:**
> "gpt-4o-audio-preview": "The audio primarily features excited human shouting and screaming. The tone is energetic and loud, suggesting a high-adrenaline situation like a thrill ride or a fun event. There is some background noise, but the main focus is the enthusiastic vocal sounds. The volume is relatively high and consistent, with a few moments of slightly varying intensity. There are no noticeable instruments or artificial effects besides natural human voices. The overall acoustic quality is lively and clear, with a sense of movement and excitement."
>
> "gemini-2.5-pro": "The audio begins with low-volume shuffling and clanking sounds. This is quickly followed by a series of high-pitched, excited human screams and yells that rapidly increase in volume. A loud, rushing sound, similar to wind, builds in the background alongside the screams. The recording ends abruptly, cutting to silence at its peak volume."
>
> ### **6. The impact of vision-based information on downstream tasks**
> In Section 6.1 of the paper, we conducted ablation experiments on different modalities and analyzed the experimental results. The experimental results show that ablating auxiliary modalities (Music, Video, Speech) generally degraded performance.

---

### Official Review · Reviewer_tyQ8 · 2025-10-30

**Soundness:** 2
**Presentation:** 3
**Contribution:** 2
**Rating:** 6
**Confidence:** 4

**Summary:**

The authors in the paper propose a two-state pipeline solution to automatically generate fine-grained audio captions.They achieve this by fusing information from multiple modalities. In first state specialized pretrained models (such as GAMA, Whisper, YamNet, OpenMu) extract complementary cues from an audio clip abd visual scene description via VLM (Qwen2.5-VL-72B) from associated video frames. A large language model (QwQ-32B) integrates these cues to generate a detailed, context-rich caption.   Using this pipeline on AudioSet (2M 10-second YouTube clips), the authors construct FusionAudio-1.2M, a dataset of 1.2 million audio clips with fine-grained captions, plus 6 million question-answer (QA) pairs generated about those clips. They also fine-tune audio-language models on this data, showing improved performance on downstream tasks

**Strengths:**

(1) The proposed pipeline in paper combines audio, speech, music, and visual cues, analogous to how humans integrate multiple senses. This yields captions with unprecedented detail and accuracy; the paper’s examples show that FusionAudio captions include context and reasoning absent in prior work.

(2) The resulting FusionAudio-1.2M dataset is a significant contribution. It is one of the largest audio-caption corpora to date (1.2M clips), and captions are substantially longer and more descriptive than those in existing datasets.

(3) Models  fine-tuned on FusionAudio show improved performance on multiple tasks.

**Weaknesses:**

(1) The method mainly relies on multiple large pretrained models (Whisper, CLAP, GAMA, OpenMu, Qwen2.5-VL, 32B LLM) in sequence. Architecture is based on pre-existing work and doesn't bring enough novelty to the paper.

(2) Although  abalation shows results for retrival task and audio understanding. The  visual dependency, should be tested using the captioning model with and without visual input. If a model trained on FusionAudio is asked to caption an audio without video, will it hallucinate visual context or perform poorly? The paper doesn’t fully explore this limitation.

(3)  The paper’s hallucination analysis focused on obvious mismatches, but more nuanced biases weren’t discussed. An important limitation is that the ground truth correctness of these detailed captions is not fully verifiable at scale some captions may include inferred context (“wind sounds suggest an outdoor environment”) that isn’t explicitly labeled as right or wrong.

(4) The paper’s evaluations, while extensive on retrieval and audio understanding tasks, did not directly evaluate caption generation quality on standard benchmarks (e.g., no BLEU, CIDEr, or SPIDEr scores on AudioCaps/Clotho test sets were reported). This could be seen as a weakness, as it’s hard to quantify how much better the fine-grained captions are in describing audio content compared to previous captions.

**Questions:**

(1) How good this captioning model work on audio-only inputs? In many real-world audio captioning applications (assistive tech, audio archives), a visual stream won’t be available.

---

> ### Author Response · Authors · 2025-11-27
>
> ### **1. The reviewer argues that our work lacks novelty, viewing it as a straightforward integration of existing models rather than a methodologically innovative contribution.**
> * **Response on Limited Novelty:**
> We respectfully argue that our main contribution lies in a methodological advance: a principled pipeline and the large-scale dataset constructed through it, both of which directly address a critical bottleneck in audio-language research—the lack of a low-hallucination, verifiable automatic data pipeline and large-scale, high-quality audio–text data. Rather than a “straightforward integration,” our system is a carefully designed architecture that aims to mimic sophisticated human perception. In particular, our two-stage caption generation pipeline is novel in how it provides diverse, structured context to the LLM for detailed synthesis, moving beyond simple feature concatenation toward complex, guided reasoning.
>
> * **Methodological Depth:**
> Our fusion strategy extends beyond which modalities to combine, to how they are synthesized. We invested significant effort in each modalities and how to effectively integrate this information module. This provides valuable and non-trivial practical insights for the large-scale scenario of using LLMs for automatically constructing data. Furthermore, our choice of a powerful LLM was deliberate. We established the necessity of using a strong reasoning model, whose advanced Chain-of-Thought (CoT) capabilities are essential for handling the complex cross-modal inference and conflict resolution required to produce higher-fidelity, less-hallucination and verifiable-process captions, which is something other related works couldn't achieve. This demonstrates the task's complexity and the sophistication of our solution.
>
> * **Novelty Demonstrated by High Data Efficiency and Impact:**
> The ultimate validation of our method's novelty is the quality and utility of the data it produces. Our models achieve state-of-the-art performance while using significantly fewer unique audio clips (Sec 5.1.2). This demonstrates that our method successfully extracts more high-quality, fine-grained information per sample—a direct result of our novel generation and synthesis pipeline and a clear step beyond prior art.
>
> In summary, while we use existing components, our novelty lies in the architecture that combines them, the sophisticated reasoning process we engineered for the LLM, and the resulting high-quality, high-efficiency data that fills a recognized gap in the field.
>
> ### **2. When a model trained on FusionAudio generates captions for audio without video, will it produce vision-related hallucinations or experience a significant drop in performance?**
>
> * The input to the model is audio only，not video.
> * The goal of our work is to use information from other modalities to assist in audio caption generation, and none of the audio captions contain visual information. If there is a conflict between visual and auditory information, the fusion model will also remain faithful to the auditory information.In Figure 3 of the paper, we present the adoption ratio of various modal information in the final captions of the dataset, where it can be seen that the usage ratio of visual information does not exceed 60%. This also validates our viewpoint—the fusion model selectively uses information from other modalities for assistance.
>
> ### 3. Fine-grained captions cannot be verified at scale.
>
> 1.The advantage of our pipeline lies in cross-validating information from different modalities and quantify the confidence via CLAP model for reference, ensuring that caption construction is evidence-based and maximizing confidence to the greatest extent.
>
> 2.To contextualize our evaluation scale, we benchmark it against prominent datasets that used similar automated annotation pipelines. Our evaluation effort is not only robust but also meets or exceeds the standard practice in our field.Meanwhile, we have also sampled a portion of data for manual validation. Compared with other related works, the validation ratio is representative and typical:
>
> | **Dataset** | **# of Captions** | **of Human-Evaluated Samples** | **Ratio(Eval/Total)** |
> | :--- | :---: | :---: | :---: |
> | WavCaps | 403k | 320 | 0.08% |
> | AudioSetCaps | 1.9M | 79 | 0.00% |
> | Auto-ACD | 1.5M | 1,200 | 0.08% |
> | FusionAudio | 1.2M | 1,300 | 0.11% |

---

> ### Author Response · Authors · 2025-11-27
>
> ### **5. The reasons for not using fixed metrics such as BLEU for evaluation.**
>
> While BLEU, CIDEr and SPIDEr on AudioCaps/Clotho are common reporting metrics, we deliberately do not treat them as our primary indicator of caption quality. Following Kothinti and Emmanouilidou (2022), who systematically evaluate BLEU, ROUGE, METEOR, CIDEr and SPICE under controlled semantic, temporal and spatial perturbations on Clotho and show that these fixed, text-only metrics often fail to reliably distinguish semantically correct captions from variants with subtle but critical errors (and sometimes even assign higher scores to worse captions)[1].
>
> We consider such overlap-based scores to be a noisy and potentially misleading signal for fine-grained, audio-grounded captioning. Our task explicitly targets detailed descriptions of sound events, sources, attributes and temporal relations, where these subtle distinctions are exactly what matters. Consequently, instead of relying on a single scalar BLEU/CIDEr/SPIDEr score, we focus on (i) **an arena-style human preference evaluation (Appendix F)**, which directly compares system outputs and better reflects perceived caption quality, and (ii) **a comparative visualization of caption embeddings across different datasets (Appendix D)**, which qualitatively illustrates how our fine-grained captions occupy a richer and more structured semantic space than existing audio captions. We believe this combination provides a more faithful and objective assessment of whether a model truly produces better, more informative audio captions than prior work.
>
> [1] Kothinti, S., & Emmanouilidou, D. (2022). Investigations in audio captioning: Addressing vocabulary imbalance and evaluating suitability of language-centric performance metrics. arXiv preprint arXiv:2211.06547.

---

### Official Review · Reviewer_pqCu · 2025-10-30

**Soundness:** 3
**Presentation:** 3
**Contribution:** 2
**Rating:** 6
**Confidence:** 4

**Summary:**

This paper introduces a method, and dataset (FusionAudio), for fine-grained audio-visual caption generation. The pipeline for audio-captioning is multimodal: it first extracts domain-specific features (general audio, speech, music presence, and visual cues), then uses an LLM to synthesize these captions into a final version (after some filtering). In this pipeline, all of the underlying components are frozen, and linked via prompt-tuning in the final LLM synthesis step. This pipeline is then used to collect an automated dataset: FusionAudio, which is the result of running this pipeline on 1.2M clips from the AudioSet dataset. The dataset is then used to train a HTSAT-BERT model for cross-modal audio/text retrieval, and it is shown that AudioFusion leads to performance improvements over existing captioning datasets. Further, the paper fine-tunes a GAMA model on FusionAudio, and shows that it helps to improve general purpose audio understanding models.

**Strengths:**

One of the most exciting areas for research right now is spoken language models (SLMs), and as a community, we are generally lacking high quality pre-training data for SLMs. This paper presents a pathway towards better SLMs, by defining a way to collect audio-visual captioning data from existing video clips. The motivation that the visual component might provide something interesting is exciting, and fairly novel (but has been explored before, for example in both [1,2] they show that visual components can help with audio understanding in ASR). The results are strong, and fairly compelling, particularly the GAMA fine-tuning results in Section 5.2. The paper is well-written, and easy to understand.

[1] Shi, Bowen, et al. "Learning audio-visual speech representation by masked multimodal cluster prediction." arXiv preprint arXiv:2201.02184 (2022).
[2] Chan, David M., et al. "Multi-modal pre-training for automated speech recognition." ICASSP 2022-2022 IEEE International Conference on Acoustics, Speech and Signal Processing (ICASSP). IEEE, 2022.

**Weaknesses:**

- It's likely that some of the improvement from FusionAudio comes from the fact that it is much larger than some of the other datasets, instead of the underlying quality of the data.  It would be good to include the number of samples in each dataset in Table 4, as well as provide a sample/compute-matched ablation in the experiments in 5.1 and 5.2 to help demonstrate that it is not just scale which is improving the performance of the downstream models.
- The pipeline isn't really trained at all, just stitched together from constituent pre-trained components. Captioning on AudioSet has already been explored (AudioSetCaps). This means that there's a bit of a lack of technical novelty: there's no particularly new strategies here (despite strong downstream improvements). The main novelty is the collection of multimodal constituent components, but it's not really ablated which makes it hard to understand which parts of the captions are leading to the marginal downstream improvements.
- The results for audio retrieval aren't particularly impressive, with AudioSet captions (a comparatively sized dataset) only performing marginally worse, and the results in Table 5 are strong compared to open models, but are not compared to closed-source models (suggesting that perhaps, a better pipeline would be to use gpt-4o-audio to collect the underlying caption set).
- There's no statistical analysis in the results. While this is mentioned in the limitations section, bootstrapping, or SEM would provide at least a glimpse of the potential downstream variance (though this is likely fairly low given the large number of test samples).
- The output captions seem somewhat prompt-tuned to appeal to the target distribution, rather than what is actually relevant to an audio caption. For example, one of the fusion prompt components is: "Speech Content.... Its specific textual content (including paraphrasing or summarization) must never appear in the final output." This will increase performance on the audio captioning task (since the actual speech content is usually not part of the caption, but I'm not sure that this is aligned with how anyone would actually want to use the system (in most cases, I think users would actually want the content of the speech to be part of the audio description).

More minor weaknesses:
- There's not much qualitative evaluation (analysis) of samples, which would help the reader to understand what kinds of samples are making up the difference in Table 4/Table 5.
- The paper claims ``fine-grained audio understanding'' but doesn't really quantify this, or demonstrate that the proposed approach is more fine-grained than any other method.

Notes (rather than weaknesses):
- I think that Table 13 (Sec. F) is actually really important to this paper, since it demonstrates that humans prefer the pipeline captions to most baseline captions, and hiding this in the appendix reduces the strength of the main point. Figure 4 doesn't really add anything to the main paper, and could be removed for this experiment.
- I don't love that the model reduces the verbalization of uncertainty (explicitly forbidding mentioning uncertainty in the prompt), since this is likely to reduce user trust in the system, and encourage more confident sounding outputs (while not necessarily being more conservative).

**Questions:**

- There's no discussion of licensing in the paper, how is the dataset licensed/will it be publicly available for research (or otherwise)?

---

> ### Author Response · Authors · 2025-11-27
>
> ### **1. The number of samples in each dataset should be included in paper. The performance of FusionAudio stems from the scale of the dataset rather than the inherent quality of the data itself.**
>
> * Amount of Datasets：The paper has clearly specified the size of each dataset in Table 2. For further clarity, we will add the dataset size information to Table 4 in subsequent revisions.
>
> * Data Scaling Ablation Study：For audio understanding task, each dataset we used to train model is aligned with the scale of 25k (Table 5). For audio retrieval task, an ablation study on data scale is conducted in Section 6.2 of the paper, and the experimental results are visualized in Figure 5.
>
> From these results, under the condition of equal data scale, the downstream task metrics of FusionAudio are consistently higher than those of other datasets.
>
> ### **2. The reviewer argues that our work lacks novelty, viewing it as a straightforward integration of existing models rather than a methodologically innovative contribution.**
> * **Response on Limited Novelty:**
> We respectfully argue that our main contribution lies in a methodological advance: a principled pipeline and the large-scale dataset constructed through it, both of which directly address a critical bottleneck in audio-language research—the lack of a low-hallucination, verifiable automatic data pipeline and large-scale, high-quality audio–text data. Rather than a “straightforward integration,” our system is a carefully designed architecture that aims to mimic sophisticated human perception. In particular, our two-stage caption generation pipeline is novel in how it provides diverse, structured context to the LLM for detailed synthesis, moving beyond simple feature concatenation toward complex, guided reasoning.
>
> * **Methodological Depth:**
> Our fusion strategy extends beyond which modalities to combine, to how they are synthesized. We invested significant effort in each modalities and how to effectively integrate this information module. This provides valuable and non-trivial practical insights for the large-scale scenario of using LLMs for automatically constructing data. Furthermore, our choice of a powerful LLM was deliberate. We established the necessity of using a strong reasoning model, whose advanced Chain-of-Thought (CoT) capabilities are essential for handling the complex cross-modal inference and conflict resolution required to produce higher-fidelity, less-hallucination and verifiable-process captions, which is something other related works couldn't achieve. This demonstrates the task's complexity and the sophistication of our solution.
>
> * **Novelty Demonstrated by High Data Efficiency and Impact:**
> The ultimate validation of our method's novelty is the quality and utility of the data it produces. Our models achieve state-of-the-art performance while using significantly fewer unique audio clips (Sec 5.1.2). This demonstrates that our method successfully extracts more high-quality, fine-grained information per sample—a direct result of our novel generation and synthesis pipeline and a clear step beyond prior art.
>
> In summary, while we use existing components, our novelty lies in the architecture that combines them, the sophisticated reasoning process we engineered for the LLM, and the resulting high-quality, high-efficiency data that fills a recognized gap in the field.
>
> ### **4. The results are not compared to closed-source models.**
>
> * **Performance Comparison with Closed-source Models**
> We compare our method with Closed-source models in audio understanding task and arena method.
>
> * **Audio Understanding Task**
> Table 5 illustrates the performance with Closed-source models in audio understanding task.It outperforms Gemini-2.5-pro/GPT-4o in adverse acoustic conditions and fine-grained information tasks — this demonstrates the value of the proposed pipeline in nuanced audio understanding. However, Gemini-2.5-pro still maintains a leading position in high-level semantic understanding (which is expected for large general-purpose models equipped with extensive world knowledge).
>
> * **Arena Method**
> We have conducted a head-to-head human evaluation using a pairwise comparison (Arena) methodology in Appendix F.1, assessing Detail and Accuracy (low hallucination). This directly tests the trade-offs between descriptive richness and factual correctness.Table 13 shows the result of Arena.It proves that on Accuracy (i.e., low hallucination), our model ranks #1, surpassing all other models.
>
> * **The reasons for not using closed-source models as pipeline modules**
> Closed-source models typically have higher memory requirements and rely on API calls. Considering cost and carbon emission concerns, we did not use them.

---

> ### Author Response · Authors · 2025-11-27
>
> ### **5. the lack of statistical analysis to quantify estimation variance in the results.**
> We implemented bootstrapping analysis across benchmarks as suggested. The results show consistent statistical precision: relative standard errors were consistently below 2% of the total scoring ranges ( most of benchmarks below 1%), with 95% confidence intervals spanning less than 10% of the possible score values across all evaluations. This uniform pattern of minimal estimation variance confirms the statistical reliability of our results, addressing your concern about potential downstream variability.
>
> ### **6. The output captions seem somewhat prompt-tuned to appeal to the target distribution, rather than what is actually relevant to an audio caption.**
> We appreciate the reviewer’s concern and we acknowledge that our current study, like most prior work, does not yet incorporate ASR content into the captions. Importantly, however, even in this simpler setting without ASR, existing audio captioning methods still exhibit clear limitations and fail to produce consistently high-quality captions. In other words, the community has not yet solved the core problem of general audio understanding and captioning without transcripts. Our work is therefore intentionally positioned as a step toward closing this gap: we first aim to significantly strengthen this transcript-free regime before moving on to more complex settings.
>
> At the same time, we fully agree that high-precision captions enriched with ASR information are valuable and often better aligned with human preferences in many downstream applications. We view such ASR-augmented captioning as a natural and important next step built on top of our framework, rather than something in conflict with our current focus. To enable this line of research, our released dataset makes all intermediate information—including ASR outputs and video captions—publicly available, so that future work can freely fuse these signals and explore more sophisticated, ASR-based captioning strategies.
>
> ### **7. Lack of qualitative cases to perceive the differences brought by the proposed method.**
> To better demonstrate the advantages of our work, we present several comparative cases with other related studies：
>
> * Case 1:
>
> "AudioSetCaps": "This audio clip features a distinct, rich bell sound, creating an evocative and resonant sonic experience.",
>
> "Auto-ACD": "Soft ambient music plays as a clock ticks, creating a serene atmosphere in an indoor setting.",
>
> "Sound-VECaps": "Music is playing softly in the background while a person works on their laptop in a peaceful home office, surrounded by plants and natural light, with occasional snorts indicating a moment of amusement or surprise.",
>
> "FusionAudio": "The audio features a continuous ringing sound resembling a bell layered over a hypnotic, whining pad guitar in a monaural recording. Intermittent bird-like chirps emerge alongside shimmering hi-hat rhythms. As the piece progresses, the music transitions into a noisier, more mysterious tone while retaining subtle peaceful elements."
>
> **Analysis:** In addition to identifying the bell sound recognized by most captions, our method further supplements bird chirps and crisp hi-hat sounds. Our captions clearly specify the instrument type as a whining pad guitar, whereas other captions merely indicate the presence of music.
>
>
> * Case 2:
>
> "AudioSetCaps": "Male child makes a sound as he interacts with a scurrying mouse, creating an engaging and lively audio clip.",
>
> "Auto-ACD": "A woman speaks in a small room, with a probability of 0.85, according to the audio tags.",
>
> "Sound-VECapsl": "A child is speaking and then holds a toy in front of their face, followed by partially obscuring their face with a black object, all while sitting in a dimly lit room with a white door and vent.",
>
> "FusionAudio": "The audio primarily features a child's clear, continuous speech with intermittent soft mechanical noises resembling a small device or toy in the room. The child's voice remains dominant throughout, while the periodic mechanical sounds appear to accompany an interactive activity involving controlled movements."
>
> **Analysis:** The core content of the audio consists of a young boy's speech and mechanical white noise, and only FusionAudio can describe them accurately.

---

> ### Author Response · Authors · 2025-11-27
>
> * Case 3:
>
> "AudioSetCaps": "A male speaker is heard in the audio, accompanied by crowd cheers and water splashing sounds. The speaker's emotion cannot be discerned from the audio alone.",
>
> "Auto-ACD": "An adult male delivers a speech while a crowd enthusiastically cheers and applauds in an auditorium.",
>
> "Sound-VECaps": "A man yelling followed by a man screaming while wind blows into a microphone, as a zip line runs across a picturesque coastal landscape with a beach, town, and rolling hills, on a sunny day.",
>
> "FusionAudio": "A dynamic auditory scene features clear human vocalizations including shouts, screams, and laughter. These are accompanied by consistent rhythmic sounds that maintain a steady tempo throughout, creating contrast between the active vocal elements and the repetitive background noise."
>
> **Analysis:** The core of the audio lies in humans' joyful shouts and the sound of the zip line, and only FusionAudio can describe them accurately. "Sound-VECaps" even add the visual information in final caption, which is unacceptable.
>
> We will put these case studys in updated version.
>
> ### **8. Lack of quantitative analysis to prove fine-grained audio understanding**
>
> We have calculated multiple metrics across different datasets, all of which can corroborate the fine-grained audio understanding capability of our dataset.
>
> * **Caption Length.** We present the comparison of subtitle length distributions across different datasets in Figure 3. Our dataset has the longest subtitles, and combined with its excellent performance on downstream tasks, we believe this is sufficient to demonstrate the gain in fine-grained audio understanding.
>
> * **Diversity t-SNE.** We utilized the text tower of the CLAP model to generate subtitles for different sound types, performed dimensionality reduction using t-SNE, and finally calculated the inter-class distance and intra-class distance of various sound types. The metrics demonstrate that our subtitles exhibit better discriminability for different sound categories. Visualizations of the experimental results are available in Appendix D and Figure 4.
>
> * **Better downstream performance** In downstream audio understanding tasks, we have dedicated benchmarks for evaluating fine-grained information capture capability. When compared with the same model trained on other datasets, our dataset achieves the best performance on these benchmarks.The result is available in Table5.
>
> ### **9. The location of Table 13 and Figure 4**
> We appreciate the feedback. In the revision, we will carefully reconsider the positioning of key figures and tables to strengthen the narrative focus and evidential support.
>
> ### **10. No uncertain information is included in the dataset.**
> Thank you for the point. Our intent with FusionAudio is to decouple content accuracy from output style. We curate a captioning corpus that minimizes stylistic variance and stays faithful to event-level facts. This choice also preserves room for subsequent augmentation/data augmentation by other following works (if needed): uncertainty expression or refusal policies, whose desirability is application-dependent (e.g., conservative Q&A and creative tasks) and lacks a single human-preference optimum, can be added in post-training via other ways, like style-specific datasets. We will clarify this scope.
>
> ### **11. The discussion of licensing**
> All models and data we used are open-source and release license online. We will clarify this in later revised version.
> Built on the open-source dataset, FusionAudio has been made open-source to the community.

---

### Official Review · Reviewer_B87y · 2025-11-01

**Soundness:** 2
**Presentation:** 3
**Contribution:** 2
**Rating:** 4
**Confidence:** 3

**Summary:**

This work proposes a pipeline for generating fine-grained audio captions, inspired by human auditory perception. The method involves using specialized expert models (for speech, music, general sound, and visual context) to extract multimodal cues, which are then synthesized by a LLM to produce detailed descriptions. The primary output is FusionAudio-1.2M, a large-scale, open-source dataset constructed using this pipeline. The authors demonstrate the utility of this dataset by training enhanced audio-text retrieval and audio understanding models, and they conduct ablations to confirm the contribution of each modality, with visual information being particularly significant.

**Strengths:**

The construction and open-sourcing of the FusionAudio-1.2M dataset is a significant contribution. Large-scale, high-quality audio-text data is a critical bottleneck in the field, and this resource will be extremely helpful for subsequent research and model development. The paper provides a thorough experimental analysis that clearly demonstrates the importance of integrating multiple modalities. The ablation study convincingly shows that each modality (especially visual context) contributes to the final model performance, offering valuable insights for the community. The evaluation is extensive, covering audio-text retrieval, audio understanding on a wide range of tasks, and even a human preference study. This multi-faceted assessment provides strong evidence for the quality of the generated captions and the models trained on them.

**Weaknesses:**

The core pipeline of "using expert models to extract features + an LLM to synthesize them" is a common and well-established paradigm for data augmentation and caption generation. While the specific application to audio and the inclusion of visual cues is valuable, the overall approach lacks fundamental innovation. As a methodological paper, it feels incremental. Also, the decision to build yet another large-scale dataset on top of AudioSet is a notable limitation. The community is already saturated with AudioSet-derived corpora, which limits diversity and can perpetuate the biases and limitations inherent in the original dataset. A more novel audio source would have significantly increased the impact. The framing of the work through "human auditory perception" feels somewhat forced and "flashy but insubstantial." The technical contributions stand on their own, and this biological analogy does not add substantial scientific rigor or insight, potentially detracting from the clear presentation of the method.

**Questions:**

1. You identified a potential mismatch between video and audio content (e.g., sounds originating from outside the frame). How does this issue impact caption quality, and did you implement any specific filtering or processing on the video information to mitigate it?
2. Given the community's concern about over-reliance on AudioSet, what was the rationale for not sourcing audio from a more diverse or novel set of videos? Are there plans to extend FusionAudio beyond AudioSet in the future?
3. The pipeline relies on multiple large, proprietary models (e.g., Qwen2.5-VL-72B, QwQ-32B). Could you discuss the computational cost and carbon footprint of generating the 1.2M captions? Is the pipeline feasible for most research groups to reproduce?
4. In the ablation study, removing speech information sometimes improved performance on Task 1. Could you elaborate on the potential reasons? Is this solely due to poor ASR quality, or could it indicate that for some tasks, paralinguistic cues are more important than the lexical content?
5. How does your LLM-based fusion method compare to a simpler, non-LLM approach (e.g., a rule-based template or a smaller, trained fusion model) in terms of cost, control, and final caption quality?

---

> ### Author Response · Authors · 2025-11-27
>
> ### **1. The reviewer argues that our work lacks novelty.**
> * **Response on Limited Novelty:**
> We respectfully argue that our main contribution lies in a methodological advance: a principled pipeline and the large-scale dataset constructed through it, both of which directly address a critical bottleneck in audio-language research—the lack of a low-hallucination, verifiable automatic data pipeline and large-scale, high-quality audio–text data. Rather than a “straightforward integration,” our system is a carefully designed architecture that aims to mimic sophisticated human perception. In particular, our two-stage caption generation pipeline is novel in how it provides diverse, structured context to the LLM for detailed synthesis, moving beyond simple feature concatenation toward complex, guided reasoning.
>
> * **Methodological Depth:**
> Our fusion strategy extends beyond which modalities to combine, to how they are synthesized. We invested significant effort in each modalities and how to effectively integrate this information module. This provides valuable and non-trivial practical insights for the large-scale scenario of using LLMs for automatically constructing data. Furthermore, our choice of a powerful LLM was deliberate. We established the necessity of using a strong reasoning model, whose advanced Chain-of-Thought (CoT) capabilities are essential for handling the complex cross-modal inference and conflict resolution required to produce higher-fidelity, less-hallucination and verifiable-process captions, which is something other related works couldn't achieve. This demonstrates the task's complexity and the sophistication of our solution.
>
> * **Novelty Demonstrated by High Data Efficiency and Impact:**
> The ultimate validation of our method's novelty is the quality and utility of the data it produces. Our models achieve state-of-the-art performance while using significantly fewer unique audio clips (Sec 5.1.2). This demonstrates that our method successfully extracts more high-quality, fine-grained information per sample—a direct result of our novel generation and synthesis pipeline and a clear step beyond prior art.
>
> In summary, while we use existing components, our novelty lies in the architecture that combines them, the sophisticated reasoning process we engineered for the LLM, and the resulting high-quality, high-efficiency data that fills a recognized gap in the field.
>
> ### **2. The impacts of conflicts between visual and audio information when fusing multi-modal data, as well as our corresponding handling methods.**
> **Our core principle is to maintain fidelity to the audio.** The primary goal is to generate audio captions, using other modalities as auxiliary context only when they can clarify or enrich the auditory information without contradicting it. Visual information is explicitly forbidden from overriding or fabricating auditory facts.
>
> This principle is enforced through a carefully designed hierarchical prompting strategy for the LLM synthesizer (detailed in Appendix C, Fig. 14-17). The prompt establishes a strict priority order:
>
> *  High-Confidence Audio Information (e.g., AudioSet tags) is paramount.
> *  Core Audio Descriptions (from unimodal models) form the next tier.
> *  Visual Information is assigned a subordinate role, permitted only to:
>     *   Clarify Auditory Ambiguity: For instance, if the audio contains a generic "rumbling sound," a visual of an airplane can refine the caption to "airplane engine."
>     *   Act as an Internal Sanity Check: If the video shows a static image while the audio contains complex sounds, the model is instructed to trust the audio and ignore the irrelevant visual stream.
>
> Crucially, the prompt explicitly instructs the model: "Video information must never be used to negate or modify known auditory facts... This inconsistency is only used as an internal decision marker, not for generating output."
>
> To demonstrate the effectiveness of this approach, we present three cases of audio-visual mismatch and our model's output (due to space constraints, the content has been condensed). In all examples, the model successfully prioritizes audio fidelity:
>
> *   Case 1: Off-screen Sound (Telephone Ringing):
>     *   Audio: "telephone ringing"
>     *   Video: "two men sitting still, no phone visible"
>     *   Output: The caption correctly describes the telephone sounds, completely and correctly ignoring the contradictory visual information.
>
> *   Case 2: Ambiguous Sound (Sonar-like Synthesizer):
>     *   Audio: "sonar-like sound," with a high-confidence tag for "Synthesizer (90%)"
>     *   Video: "desolate landscape, no water"
>     *   Output: The model uses the video to disambiguate the audio description. Seeing no water, it correctly trusts the "Synthesizer" tag and refines the caption to "continuous synthesized electronic tones resembling sonar pings," avoiding the incorrect "submarine" hallucination. This is a perfect example of the intended synergy.

---

> ### Author Response · Authors · 2025-11-27
>
> *   Case 3: Irrelevant Visuals (Static Title Card):
>     *   Audio: "Electronic music, techno and trance"
>     *   Video: "Static title card with text"
>     *   Output: The caption provides a rich description of the music, correctly identifying the video as irrelevant and excluding any mention of the title card, text, or colors.
>
> These examples confirm that our system is robust to audio-visual mismatches by design. It does not require a separate filtering step for this issue because the logic is embedded directly into the LLM's synthesis instructions.
>
>
> ### **3. The rationale for not constructing data from other datasets, given that numerous studies have been conducted on AudioSet within the research community.**
>
> Our choice to use AudioSet was a strategic one, designed to provide the clearest possible validation of our paper's central contribution: the multimodal contextual fusion pipeline.
>
> *   Isolating the Methodological Contribution: The primary novelty of our work is the two-stage pipeline for generating fine-grained captions (Sec. 3.1, Fig. 2), inspired by human auditory perception. To rigorously evaluate the efficacy of this *pipeline*, it was crucial to control for other variables. Using AudioSet, the de facto standard for many recent audio captioning works (e.g., AudioSetCaps, Auto-ACD, see Table 2), allows for a direct and fair comparison. This ensures that the superior performance of our resulting models in downstream tasks (Tables 4 & 5) and the richer semantic structure of our dataset (Fig. 4 & Table 11) can be unambiguously attributed to our proposed method, rather than differences in the source video/audio characteristics.
>
> *   Maximizing Community Impact and Demonstrating Scalability: By applying our method to AudioSet, we demonstrate its immediate value and scalability. We show how our pipeline can take a ubiquitous, foundational dataset and significantly "upgrade" it, producing the far more detailed and context-aware FusionAudio-1.2M. This provides a direct and tangible benefit to the research community, which is already heavily invested in this data source.
>
> *   Future Work: We fully agree with the reviewer's sentiment that the community needs more diverse datasets. Our work provides the very tool to create them efficiently. We plan to extend our validated pipeline to additional audio–video corpora (e.g., VGGSound) to further broaden the landscape of high-quality audio–language data. Moreover, we intend to maintain our dataset as a continuously updated open-source resource, where newly processed samples, improved annotations, and extended modalities can be incrementally incorporated. This will allow the community to benefit from a living, evolving dataset rather than a static snapshot, and to build upon our framework for increasingly comprehensive audio understanding research.
>
> ### **4. The Computational Cost and Reproducibility of the FusionAudio Pipeline.**
> #### **4.1 On Computational Cost & Carbon Footprint:**
>
> We agree that transparency regarding computational expenditure is crucial. The primary cost is a one-time effort to generate the **FusionAudio-1.2M** dataset, which we are releasing publicly to benefit the entire research community. We have broken this down for clarity:
>
> *   **Specialist Models (One-time pass over the dataset):**
>     *   GAMA: ~70 A40 GPU-days
>     *   Whisper: ~20 A40 GPU-days
>     *   OpenMu: ~24 A40 GPU-days
> *   **LLM/VLM Integrators (Per-sample token cost):**
>     *   Qwen2.5-VL-72B (Visual): ~10.2k tokens/sample
>     *   QwQ-32B (Fusion & QA): ~9.3k tokens/sample
>
> While this represents a significant one-time computational investment, we argue that it is a necessary and efficient one. By creating and sharing this high-quality, fine-grained dataset, we amortize this cost across the many research groups who can now use **FusionAudio-1.2M** to train and evaluate more efficient models without needing to reproduce the generation pipeline themselves. We will add a detailed breakdown and discussion of these costs in the camera-ready appendix to ensure full transparency.
>
> #### **4.2 On Reproducibility & Feasibility:**
>
> We would like to respectfully correct a key point in the review: **all models used in our pipeline are publicly accessible, not proprietary.**
>
> *   **Model Accessibility:** As detailed in Section 3.1, our pipeline is constructed from well-known, open-source, or publicly available models. We will emphasize the public availability of these components more explicitly to prevent any ambiguity.
>
> *   **Pipeline Open-Sourcing:** To further maximize reproducibility and community benefit, **we commit to open-sourcing our complete data generation pipeline code**, including all processing scripts and the specific prompts (Appendix C) used for each model. This will allow other researchers to fully inspect, validate, and adapt our methodology.

---

> ### Author Response · Authors · 2025-11-27
>
> With this in mind, we distinguish between two aspects of reproducibility:
>
> *   **Methodological Reproducibility:** Our core contribution is a modular and transparent *method*. With our open-sourced code and the use of public models, any research group can replicate our pipeline's logic, verify its effectiveness on a smaller data subset (as we did for our 25k-sample ablations in Sec. 6.1), or apply it to new data.
>
> *   **Large-Scale Dataset Generation:** While re-generating the entire 1.2M-sample dataset is computationally intensive, this is precisely the contribution we aim to provide *as a service* to the community. By undertaking this one-time effort and releasing **FusionAudio-1.2M**, we democratize access to fine-grained audio data and enable researchers to bypass the generation cost entirely. Our experiments (Table 5) show that even small, high-quality subsets of our data can yield state-of-the-art results, demonstrating its immediate value and data efficiency.
>
> We will update our Reproducibility Statement to reflect our commitment to releasing the pipeline code and to clarify the public nature of all constituent models.
>
> ### **5. On why “reducing speech cues” can improve Task 1**
> We believe the effect is not solely due to ASR quality, but primarily due to task-signal alignment in Task 1. The labels are driven by paralinguistic and non-speech evidence (background events, textures, onsets/offsets/overlaps under interference), while lexical content is only weakly predictive for the semantic level. In this setting, down-weighting explicit lexical cues tends to focus model capacity/attention on spectro-temporal feature that actually determine the label. We will clarify this task-dependence in the paper.
>
> ### **6. FusionAudio's performance in terms of cost, controllability, and final caption quality.**
>
> The core of our contribution lies in the LLM's unique ability to perform sophisticated **contextual synthesis and reasoning**, which goes far beyond the capabilities of rule-based templates or smaller, specialized models. Specifically, an LLM-based system provides two key benefits that are critical for generating *fine-grained* and *accurate* captions:
>
> *  **Conflict Resolution and Disambiguation:** As shown in our paper (Sec 3.1, Stage 2), the LLM acts as an "integration engine." It doesn't just aggregate inputs; it resolves inconsistencies between the outputs of the specialist models (e.g., when visual cues from Qwen2.5-VL-72B contradict an initial audio event from GAMA). A rule-based system would struggle with such nuanced but significant conflicts, while our LLM can infer the most plausible interpretation based on the complete multimodal context.
>
> *  **Rich, Diverse, and Natural Language Generation:** Rule-based or template-based methods inherently produce repetitive and stylistically limited captions. In contrast, the LLM leverages its vast pre-training to generate descriptions that are not only detailed but also linguistically diverse and natural, as evidenced by the significantly longer and more descriptive captions in our FusionAudio dataset (Fig 3b, Table 2). This richness is essential for training more capable models with stronger capability of instruction following.
>
> To empirically validate the cost-performance trade-off you mentioned, we have added a **new ablation study** specifically targeting the computational cost of our pipeline components. We replaced each of our high-performance "expert" models with a smaller, more efficient alternative and measured the impact on a downstream retrieval task.
>
> **Table: Ablation on Model Scale and Computational Cost. We show the performance on Audio-Text Retrieval (AudioCaps test set) after replacing expert models with smaller variants.**
>
> | **Configuration** | **T-A R@1** | **A-T R@1** |
> | :--- | :---: | :---: |
> | **Full Pipeline (Ours)** | **39.7** | **49.7** |
> | Replace GAMA -> LTU | 38.1 (-1.6) | 46.8 (-2.9) |
> | Replace Qwen2.5-VL-72B -> 7B | 38.5 (-1.2) | 47.8 (-1.9) |
> | Replace QwQ-32B (Fusor) -> Qwen-32B | 38.6 (-1.1) | 49.6 (-0.1) |
> | Replace Whisper -> Whisper-tiny | 37.5 (-2.2) | 49.7 (+0.0) |
>
> As the results demonstrate, while using smaller models does reduce performance, the pipeline remains effective. It shows that our pipeline is not rigidly dependent on massive models. Instead, it provides a flexible framework where researchers can **balance performance and computational cost** to suit their specific needs, choosing the optimal operating point on the cost-effectiveness curve.
>
> We will add this new ablation study and a detailed discussion to the revised version to make this trade-off explicit. Thank you again for prompting this valuable addition.

---

### Note · Authors · 2026-01-07

I have read and agree with the venue's withdrawal policy on behalf of myself and my co-authors.